# Subclass-balancing Contrastive Learning for Long-tailed Recognition

## Abstract

Long-tailed recognition with imbalanced class distribution naturally emerges in practical machine learning applications. Existing methods such as data reweighing, resampling, and supervised contrastive learning enforce the class balance with a price of introducing imbalance between instances of head class and tail class, which may ignore the underlying rich semantic substructures of the former and exaggerate the biases in the latter. We overcome these drawbacks by a novel "subclass-balancing contrastive learning (SBCL)" approach that clusters each head class into multiple subclasses of similar sizes as the tail classes and enforce representations to capture the two-layer class hierarchy between the original classes and their subclasses. Since the clustering is conducted in the representation space and updated during the course of training, the subclass labels preserve the semantic substructures of head classes. Meanwhile, it does not overemphasize tail class samples so each individual instance contribute to the representation learning equally. Hence, our method achieves both the instance- and subclass-balance, while the original class labels are also learned through contrastive learning among subclasses from different classes. We evaluate SBCL over a list of long-tailed benchmark datasets and it achieves the state-of-the-art performance. In addition, we present extensive analyses and ablation studies of SBCL to verify its advantages.

## 1 Introduction

In reality, the datasets often follow the Zipfian distribution over classes with a long tail (Zipf, 2013; Spain & Perona, 2007), $i.e.$, a few classes (head classes) containing significantly more instances than the remaining tail classes. Such tail classes could be of great importance for high-stake applications, $e.g.$, patient class in medical diagnosis or accident class in autonomous driving (Cao et al., 2019; Shen et al., 2015). However, training on such class-imbalanced datasets can result in a severely biased model with noticeable performance drop in classification tasks (Wang et al., 2017; Mahajan et al., 2018; Zhong et al., 2019; Ando & Huang, 2017; Buda et al., 2018; Collobert et al., 2008; Yang et al., 2019).

To overcome the challenges posed by long-tailed data, data resampling (Ando & Huang, 2017; Buda et al., 2018; Chawla et al., 2002; Shen et al., 2016) and loss reweighing (Byrd & Lipton, 2019; Cao et al., 2019; Cui et al., 2019; Dong et al., 2018) have been widely applied but they cannot fully leverage all the head-class samples. Very recent work discovered that supervised contrastive learning (SCL) (Khosla et al., 2020) can achieve state-of-the-art (SOTA) performance on benchmark datasets of long-tailed recognition (Kang et al., 2020; Li et al., 2022). Specifically, the $k$-positive contrastive learning (KCL) (Kang et al., 2020) and its subsequent work targeted supervised contrastive learning (TSC) (Li et al., 2022) revamp SCL by encouraging the learned feature space to be class-balanced and uniformly distributed. However, those methods enforcing class-balance often come with a price of instance-imbalance, $i.e.$, each individual instance of tail classes would have much greater impact on model training than that of head classes.

Such instance-imbalance can result in significant degradation of the performance on long-tailed recognition for several reasons. On the one hand, the limited samples in each tail class might not be sufficiently representative of the whole class. So even a small bias of them can be enormously exaggerated by class-balancing methods and result in sub-optimal learning of classifiers or representations. On the other hand, head classes usually have more complicated semantic substructures,

e.g., multiple high-density regions of the data distribution, so simply downweighing samples of head classes and treating them equally can easily lose critical structural information. For example, images of a head class "cat" might be highly diverse in breeds and colors, which need to be captured by different features but downweighing or subsampling them may easily lose such information, while a tail class "platypus" might only contain a few similar images that are unlikely to cover all the representative features. Therefore, it is non-trivial to enforce both class-balance and instance-balance simultaneously in the same method.

Can we remove the negative impact of class-imbalance while still retain the advantages of instance-balance? In this paper, we achieve both through subclass-balancing contrastive learning (SBCL), a novel supervised contrastive learning defined on subclasses, which are the clusters within each head class, have comparable size as tail classes, and are adaptively updated during the training. Instead of sacrificing instance-balance for class-balance, our method achieves both instance- and subclass-balance by exploring the head-class structure in the learned representation space of the model-in-training. In particular, we propose a *bi-granularity contrastive loss* that enforces a sample (1) to be closer to samples from the same subclass than all the other samples; and (2) to be closer to samples from a different subclass but the same class than samples from any other subclasses. While the former learns representations with balanced and compact subclasses, the latter preserves the class structure on subclass level by encouraging the same class's subslasses to be closer to each other than to any different class's subclasses. Hence, it can learn an accurate classifier distinguishing original classes while enjoy both the instance- and subclass-balance.

In this paper, we apply SBCL for several visual recognition tasks to demonstrate SBCL superiority over other previous works (*e.g.*, KCL(Kang et al., 2020),TSC (Li et al., 2022)). To summarize, this paper makes the following contributions:

(a). We provide a new design principal of leveraging supervised contrastive learning for long-tailed recognition, *i.e.*, aiming at achieving both instance- and subclass-balance instead of class-balance at the expense of instance-balance.

(b). We propose a novel instantiation of the aforementioned design principal, subclass-balancing contrastive learning (SBCL), which consists of two major components, namely, subclass-balancing adaptive clustering and bi-granularity contrastive loss.

(c). Empirically, we compare the SBCL against state-of-the-art methods on three visual tasks: image classification, object detection, and instance segmentation to demonstrate its effectiveness on handling class imbalance. We also conduct a series of experiments to analyze the efficacy of SBCL.

## 2 BACKGROUND AND NOTATIONS

**Long-tailed recognition.** Long-tailed recognition aims to learn a classifier from a training dataset with long-tailed class distribution, *i.e.*, a few classes contain many data (head classes) while most classes contain only a few data (tail classes), where the major challenge is to require model recognizing all classes equally well. Let $\mathcal{D} = \{x_i, y_i\}_{i \in [n]}$ be a long-tailed training dataset, where $x_i$ denotes a sample and $y_i \in [C]$ denotes its label. Denote by $\mathcal{D}_k \subseteq \mathcal{D}$ the set of instances belonging to class $k$ and $n_k = |\mathcal{D}_k|$ the number of samples in class $k$ The total number of training samples over $C$ classes is $n = \sum_{k=1}^{C} n_k$. Without loss of generality, we follow prior work (Kang et al., 2019; Hong et al., 2021) to assume that the classes are sorted by cardinality in decreasing order (*i.e.*, if $i < j$, then $n_i \geq n_j$), and $n_1 \gg n_C$. In addition, we define the imbalance ratio as $\max_{k \in [C]}(n_k) / \min_{k \in [C]}(n_k) = n_1/n_C$. Finally, let $f_\theta(\cdot)$ be a deep feature extractor, *e.g.*, a neural network, parameterized by $\theta$ and $\mathbf{w}_c$ is the linear classifier of class $c$, then the classifier we aim to learn is $h(x_i) = \arg \max_{c \in [C]} \mathbf{w}_c^\top f_\theta(x_i)$.

**Supervised contrastive learning.** Recent studies have shown that supervised contrastive learning (SCL) (Khosla et al., 2020) provides a strong performance gain for long-tailed recognition and its variants have achieved state-of-the-art (SOTA) performance (Kang et al., 2020; Li et al., 2022). Specifically, SCL learns the feature extractor $f_\theta(\cdot)$ via maximizing the discriminativeness of positive instances, *i.e.*, instances from the same class, and the learning objective for a single training data

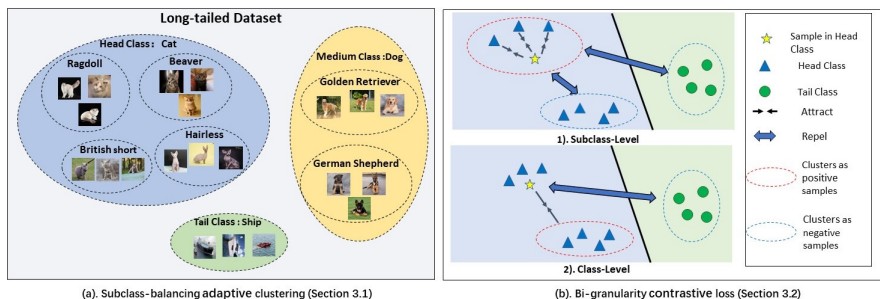

Figure 1: Illustration of subclass-balancing contrastive learning (SBCL).

$(x_i, y_i)$ in a batch $\mathcal{B} = \{(x_i, y_i)\}_{i \in [N]}$, is

$$\mathcal{L}_{SCL} = \sum_{i=1}^{N} -\frac{1}{|\tilde{P}_i|} \sum_{z_p \in \tilde{P}_i} \log \frac{\exp(z_i \cdot z_p^{\top}/\tau)}{\sum_{z_a \in \tilde{V}_i} \exp(z_i \cdot z_a^{\top}/\tau)}, \tag{1}$$

where $\tau$ is the temperature hyperparameter, $z_i = f_{\theta}(x_i)$ is the feature generated from $x_i$, $V_i = \{z_i\}_{i \in [N]} \setminus \{z_i\}$ is the current batch of features except for $z_i$, $P_i = \{z_j \in V_i : y_j = y_i\}$ is a set of instances with the same label as $x_i$. Finally, let $\tilde{z}_i$ be the feature of $\tilde{x}_i$, the augmented version of $x_i$, and for any set $S_i$ indexed by $i$, we use $\tilde{S}_i = S_i \cup \{\tilde{z}_i\}$, e.g., $\tilde{V}_i = V_i \cup \{\tilde{z}_i\}$. However, for the long-tailed datasets, the feature spaces is dominated by head classes and thus have limited capability of semantic discrimination (Kang et al., 2020). To address this, the $k$-positive contrastive learning (KCL) (Kang et al., 2020) attempts to balance the feature space by keeping the number of positive instances in $\tilde{P}_i$ equal for each class, leading to the following loss

$$\mathcal{L}_{KCL} = \sum_{i=1}^{N} -\frac{1}{k+1} \sum_{z_p \in \tilde{P}_i^k} \log \frac{\exp(z_i \cdot z_p^{\top}/\tau)}{\sum_{z_a \in \tilde{V}_i} \exp(z_i \cdot z_a^{\top}/\tau)} \tag{2}$$

where $P_i^k$ is a subset of $P_i$ with $k$ randomly drawn instances. Finally, the learned feature extractor $f_{\theta}(\cdot)$ is exploited in a sequel stage of training the classifier for long-tailed recognition (Kang et al., 2020; Li et al., 2022).

## 3 METHODOLOGY

As mentioned above, KCL and its sequels (Kang et al., 2020; Li et al., 2022) balance the learning objective of SCL by picking the same number of positive instance for each class, i.e., $|P_i^k| = k$ in Eq. 2 no matter which class $x_i$ belongs to, however, we argue that such a class-balancing approach would inevitably introduce instance-imbalance: the instances of tail classes have much more chances to be engaged in the training than that of head class. Specifically, assume each class has no less than $k$ instances, then the probability of an instance of class $c$ being selected as positive instance is $p(c) = \frac{k}{n_c}$; if the tail class $n_C$ has only $k$ instances and the imbalance ratio is $\frac{n_1}{n_C} = 100$, then we have $p(1) = \frac{k}{100k} = 0.01$ while $p(C) = 1$. We can see that when the instances of head class are selected once, that of tail class may already be trained 100 times. Thus, the training is immensely biased towards the few samples in each tail class. Besides, as tail classes only have very few instances that are not necessarily representative, the learned feature space might be unsatisfactory and sensitive to the training data of tail classes.

Here, we provide a new prospective of handling class-imbalance issue by contrastive learning: instead of aiming at class-balance at the expense of instance-imbalance, we propose to achieve both instance- and subclass-balance. We argue that head classes typically contain more diverse instances and thus have richer semantics in the training dataset. Therefore, it might be wise to break down the head classes into multiple semantically coherent subclasses, each of which consists of similar number of instance as tail classes. Built on this spirit, we develop subclass-balancing contrastive learning (SBCL), a new contrastive learning framework for long-tailed recognition (visualized in Figure 1) that achieves both instance- and subclass-balance.

---

**Algorithm 1** Subclass-balancing Adaptive Clustering

---

**Require:** Sample set $\mathcal{S} = \{x_i\}_{i=1}^n$; A threshold $M$; The number of iterations $K$;
**Ensure:** Cluster assignments for samples in $\mathcal{S}$

    **for** $k = 0$ to $K$ **do**
        Update the cluster centers $y_j = \frac{1}{n_i} \sum_{i=1}^{n_i} x_i$.         ▷ $n_i$ is the number of samples in a cluster
        Construct the cluster center set $\mathcal{C} = \{y_j\}_{j=1}^m$.         ▷ $m$ is the number of cluster centers
        **while** $\mathcal{S} \neq \phi$ **do**         ▷ Assign samples to centers $y_i$
          Select the most similar pair $(x_i, y_j) = \underset{x \in \mathcal{S}, y \in \mathcal{C}}{\arg\max} \text{ cosine-similarity}(x, y)$.
          Assign the sample $x_i$ to the center $y_j$.
          Delete the assigned sample $x_i$ from the sample set $\mathcal{S} = \mathcal{S}/\{x_i\}$.
          **if** $n_i \geq M$ **then**         ▷ Sample number in a cluster exceeds the threshold $M$
            Delete the cluster center $y_j$ from the cluster center set $\mathcal{C} = \mathcal{C}/\{y_j\}$.
          **end if**
        **end while**
    **end for**

---

### 3.1 SUBCLASS-BALANCING ADAPTIVE CLUSTERING

We break down the class into several "subclasses" to attack the imbalanced phenomenon. Particularly, given a class $c$ and the associated set of data $\mathcal{D}_c$, we employ a clustering algorithm of choice based on the features extracted by current feature extractor $f_\theta(\cdot)$ to divide $\mathcal{D}_c$ into $m_c$ subclasses/clusters. We use $\Gamma_c(x_i)$ to denote the cluster label of an instance $x_i$ of class $c$. To ensure that the number of samples for each subclass is roughly the same, we propose a new cluster algorithm to divide the unit-length feature vectors, *i.e.*, the features output by $f_\theta(\cdot)$ with additional unit-length normalization. The new proposed cluster algorithm is described in Algorithm 1. In the process of assigning vectors to their centers, we set a threshold $M$ as an upper limit of sample size in a cluster, which guarantees clusters of balanced sizes. We show the distribution of the sample size in clusters on the benchmark dataset in Appendix A.1. Specifically, the threshold $M$ is

$$M = \max(n_C, \delta) \tag{3}$$

where $n_C$ is the size of tail class and the hyperparameter $\delta$ controls the lower bound of sample size in clusters to prevent overly-small cluster. Note that we only apply clustering algorithm to classes which contain multiple instances while the tail classes remain unchanged. As a consequence, the size of each resultant cluster is similar to that of tail class $n_C$. In addition, instead of only clustering once at the beginning, we update the cluster assignment adaptively based on the current feature extractor $f_\theta(\cdot)$ during training process and empirically show that such adaptive clustering outperforms only-once clustering in Section 4.5.

Then, by replacing the class labels of head classes used in SCL/KCL with the finer-grained cluster labels, we ensure the instance-balanced, *i.e.*, each instance has similar chance of being selected regardless of its class. By breaking down the head classes, which typically contain more diverse instances, into multiple semantically coherent subclasses, we achieve subclass-balanced (instead of class-balanced) while maintain the rich semantics rendered by head classes in training dataset.

### 3.2 BI-GRANULARITY CONTRASTIVE LOSS

We now have two types of label for instances in head classes from different granularities: the coarse-grained *class* label and the fine-grained *cluster* label. A direct consequence of replacing class label in SCL/KCL with cluster label is that we no longer distinguish instances from different head classes, and therefore the boundaries between classes might be blurry, leading to optimal feature space. As a remedy, we combine the contrastive loss of both class label and cluster label into the following one and reuse the notations of Eq. 1:

$$\mathcal{L}_{SBCL} = -\sum_{i=1}^N \Big( \frac{1}{|\tilde{M}_i|} \sum_{z_p \in \tilde{M}_i} \log \frac{\exp(z_i \cdot z_p^\top / \tau_1)}{\sum_{z_a \in \tilde{V}_i} \exp(z_i \cdot z_a^\top / \tau_1)}$$

$$+ \beta \frac{1}{|\tilde{P}_i| - |M_i|} \sum_{z_p \in \tilde{P}_i / M_i} \log \frac{\exp(z_i \cdot z_p^\top / \tau_2)}{\sum_{z_a \in \tilde{V}_i / M_i} \exp(z_i \cdot z_a^\top / \tau_2)} \Big) \tag{4}$$

where $M_i = \{z_j \in P_i : \Gamma_{y_i}(x_i) = \Gamma_{y_i}(x_j)\}$ is a set of instances with the same cluster label as $x_i$. $\beta$ is a hyperparameter that balances these two loss terms. The first term corresponds to the SCL loss with cluster labels, while the second term leverages the class label but does not consider the instances of the same cluster, *i.e.*, the instances in $M_i$ are removed in the second term. Such a design choice reflects the two types of positive instances for $z_i$: (1) the instance in the same cluster and (2) the instance of the same class but in different clusters.

According to previous studies (Wang & Liu, 2021; Hoffmann et al., 2022; Li et al., 2021a), the temperature $\tau$ in contrastive loss is critical in controlling the local separation and global uniformity of the feature distribution. Specifically, a low temperature forces the features to concentrate, while as the temperature increases, the features would distribute more uniformly. Although the above objective explicitly considers the two types of label from different granularities, it still treats class and cluster label similarly. Intuitively, we expect instances of the same subclass to form a more concentrated cluster in feature space than those of the same class, since subclass naturally indicates finer-grained semantic coherence. To achieve this, we ensure the temperature $\tau_2 > \tau_1$ and dynamically adjust $\tau_2$ for each class according to its current level of concentration of the instances' feature. Following (Li et al., 2021a), for class $c$ we define $\phi(c)$ as

$$\phi(c) = \frac{\sum_{i=1}^{n_c} \|z_i - t_c\|_2}{n_c \log(n_c + \alpha)}, \tag{5}$$

where $t_c$ is the centroid for the class $c$, $\alpha$ is a hyperparameter to ensure that $\phi(c)$ is not overly-large, and $z_i$ corresponds to instances of class $c$. From the formulation, we can see that if the current averaged distance to the class centroid is large or the class contains fewer data, thus the temperature will be set large to adopt the feature distribution of class $c$ during the training process. Then we define the temperature of class $c$ as

$$\tau_2(c) = \tau_1 \cdot \exp\left(\frac{\phi(c)}{\frac{1}{C}\sum_{i=1}^{C}\phi(i)}\right) \tag{6}$$

such that $\tau_2(c)$ for class label is always larger than $\tau_1$ for cluster label (since $\phi(c) > 0$) and could reflect the current level of concentration of the instances in a class. In particular, the proposed $\tau_2(c)$ encourages the features of instances in class $c$ to form a less tight cluster than that of a subclass (by $\tau_2(c) > \tau_1$) while adaptively adjust the temperature to prevent an overly-loose/dense cluster.

### 3.3 TRAINING ALGORITHM

Here, we describe the overall training process of subclass-balancing contrastive learning and the algorithm can be found in Algorithm 2. First, the adaptive clustering (Section 3.1) could be noisy at the early stage of training (Li et al., 2021a; Wang et al., 2021b). Thus, we warm-up the feature extractor $f_\theta(\cdot)$ by a few epochs of training on ordinary SCL or KCL loss. In addition, our algorithm involves two adaptively-adjusting parts, namely, the cluster assignment and the temperature $\tau_2(c)$ for each head class $c$. Instead of updating these every epoch, we use a hyperparameter $K$ as the update interval, *i.e.*, we update the cluster assignment and the temperature based on the current learned $f_\theta(\cdot)$ every $K$ epoch.

---

**Algorithm 2** Training Algorithm

---

**Require:** Dataset $\mathcal{D} = \{x_i, y_i\}_{i \in [n]}$; ; The update interval of cluster assignment $K$; The number of warm-up epoch $T_0$; The total number of epoch $T$; The hyperparameters $\beta$ and $\delta$.
**Ensure:** A trained feature extractor $f_\theta(\cdot)$
 1: Initialize the model parameters $\theta$
 2: Train $f_\theta(\cdot)$ with SCL/KCL for $T_0$ epochs          ▷ Warm-up stage
 3: **for** $t = T_0$ to $T$ **do**
 4:   **if** $t//K == 0$ **then**      ▷ Update cluster assignment and termperature
 5:     Update the cluster assignment based on the current feature extractor $f_\theta(x)$
 6:     Update the temperture $\tau_2$ for each head class using Eq. 5 and Eq. 6
 7:   **end if**
 8:   Train $f_\theta(\cdot)$ using Eq. 4       ▷ Subclass-balancing contrastive learning
 9: **end for**

---

## 4 EXPERIMENT

### 4.1 EXPERIMENTAL SETUP

**Datasets.** We consider three commonly used long-tailed recognition benchmark datasets: CIFAR-100-LT (Cao et al., 2019), ImageNet-LT (Liu et al., 2019), and iNaturalist 2018 (Van Horn et al., 2018). The CIFAR-100-LT and ImageNet-LT datasets are artificially generated long-tailed datasets from the class-balanced datasets (Krizhevsky et al., 2009; Russakovsky et al., 2015), and the iNaturalist 2018 dataset is a large-scale real-world dataset that exhibits long-tailed imbalance.

**Baselines.** We consider baseline methods of the following three categories: (1) class-balancing classifiers, including $\tau$-norm, LWS and cRT (Kang et al., 2019), which fixes the representation which trained by cross-entropy loss and trains the classifier with class-balanced sampling; (2) one-stage balancing loss, including CB loss (Cui et al., 2019), Focal loss (Lin et al., 2017), and LDAM loss (Cao et al., 2019). These supervised distribution-aware loss makes the model to pay more attention on the minority class during training. (3) contrastive learning methods, including SCL (Khosla et al., 2020), KCL (Kang et al., 2020), SwAV (Caron et al., 2020), PCL (Li et al., 2021a) and TSC (Li et al., 2022) which train a feature extractor with the contrastive loss and then learn a classifier given the trained feature extractor.

**Evaluation protocol.** Following (Kang et al., 2020; 2019; Li et al., 2022), we implement SBCL, as well as other contrastive learning methods, in a two-stage framework. In the first stage, we train the feature extractor with a contrastive learning method, while in the second stage, we train a linear classifier on top of the learned representation. Specifically, for CIFAR-100-LT dataset, the linear classifier is trained with LDAM loss and class re-weighting (Cao et al., 2019). For ImageNet-LT and iNaturalist 2018 datasets, the linear classifier is trained with CE loss and class-balanced sampling (Kang et al., 2019). All results are averaged over 5 trials with different random seeds. We mainly report the overall top-1 accuracy. For the two large datasets, ImageNet-LT and iNaturalist 2018 datasets, following the previous work (Liu et al., 2019), we also report the accuracy of three disjoint subsets: Many-shot classes (classes with more than 100 samples), Medium-shot classes (classes with 20 to 100 samples), and Few-shot classes (classes under 20 samples). We leave the implementation details and additional experiments to the appendix.

### 4.2 MAIN RESULTS

Table 1: **Performance comparison on ImageNet-LT and iNaturalist 2018 datasets.** Top-1 accuracy of ResNet-50 (He et al., 2016) is reported. The "Many", "Medium" ,"Few" and "All" denotes different groups.

| Backbone | ImageNet-LT | | | | iNaturalist 2018 | | | |
|---|---|---|---|---|---|---|---|---|
| Methods | Many | Medium | Few | All | Many | Medium | Few | All |
| CE | 64.0 | 33.8 | 5.8 | 41.6 | 72.2 | 63.0 | 57.2 | 61.7 |
| Focal loss (Lin et al., 2017) | 51.0 | 40.8 | 20.8 | 43.7 | - | - | - | - |
| CB-Focal (Cui et al., 2019) | - | - | - | - | - | - | - | 61.1 |
| LDAM-DRW (Cao et al., 2019) | 60.4 | 46.9 | 30.7 | 49.8 | - | - | - | 64.6 |
| OLTR (Liu et al., 2019) | 35.8 | 32.3 | 21.5 | 32.2 | 59.0 | 64.1 | 64.9 | 63.9 |
| $\tau$-norm (Kang et al., 2019) | 56.6 | 44.2 | 27.4 | 46.7 | 71.1 | 68.9 | 69.3 | 69.3 |
| cRT (Kang et al., 2019) | 58.8 | 44.0 | 26.1 | 47.3 | 73.2 | 68.8 | 66.1 | 68.2 |
| LWS (Kang et al., 2019) | 57.1 | 45.2 | 29.3 | 47.7 | 71.0 | 69.8 | 68.8 | 69.5 |
| PCL (Li et al., 2021a) | 34.7 | 26.1 | 12.3 | 27.5 | 48.5 | 45.9 | 41.7 | 44.5 |
| SwAV (Caron et al., 2020) | 37.5 | 28.3 | 15.6 | 30.1 | 51.9 | 48.4 | 43.7 | 47.0 |
| BYOL (Grill et al., 2020) | 37.7 | 28.9 | 16.3 | 30.6 | 52.3 | 48.6 | 44.1 | 47.2 |
| SCL (Khosla et al., 2020) | 61.4 | 47.0 | 28.2 | 49.8 | - | - | - | 66.4 |
| KCL (Kang et al., 2020) | 62.4 | 49.0 | 29.5 | 51.5 | - | - | - | 68.6 |
| TSC (Li et al., 2022) | 63.5 | 49.7 | 30.4 | 52.4 | 72.6 | 70.6 | 67.8 | 69.7 |
| SBCL | **63.8**±0.3 | **51.3**±0.3 | **31.2**±0.4 | **53.4**±0.3 | **73.3**±0.2 | **71.9**±0.3 | **68.6**±0.3 | **70.8**±0.3 |

The results on ImageNet-LT and iNaturalist 2018 are in Table 1. We can see that SBCL outperforms the baselines with a large margin over the two datasets. In addition, on iNaturalist 2018 dataset, SBCL outperforms the previous SOTA method by 0.7% on Many, 1.3% on Medium, 0.8% on Few and 1.1% on All, which shows the effectiveness of the proposed method in solving real-world long-tailed recognition problems such as natural species classification. Besides, SBCL is also better than existing contrastive learning method like KCL and TSC for all class splits, which demonstrates the effectiveness of the design principal of pursuing both instances- and subclass-balance in contrastive

Table 2: **Performance comparison on CIFAR-100-LT.** Top-1 accuracy of the ResNet-32 (He et al., 2016) under different imbalance ratios is reported. We also report the accuracy of our re-implemented important baselines (†) in same setting on CIFAR-100-LT. The columns of "Statistic (IR 100)" are results of different disjoint subsets on CIFAR-100-LT with imbalance ratio being 100.

| Method | Imbalance Ratio | | | Statistic (IR 100) | | |
|---|---|---|---|---|---|---|
| | 100 | 50 | 10 | Many | Medium | Few |
| CE | 38.3 | 43.9 | 55.7 | 65.2 | 37.1 | 9.1 |
| CB-CE (Cui et al., 2019) | 38.6 | 44.6 | 57.1 | - | - | - |
| Focal Loss (Lin et al., 2017) | 38.4 | 44.3 | 55.8 | 65.3 | 38.4 | 8.1 |
| CB-Focal (Cui et al., 2019) | 38.7 | 45.2 | 58.0 | 65.0 | 37.6 | 10.3 |
| CE-DRW (Cao et al., 2019) | 41.4 | 45.3 | 58.1 | - | - | - |
| CE-DRS (Cao et al., 2019) | 41.6 | 45.5 | 58.1 | - | - | - |
| LDAM (Cao et al., 2019) | 39.6 | 45.0 | 56.9 | - | - | - |
| LDAM-DRW (Cao et al., 2019) | 42.0 | 46.6 | 58.7 | 61.5 | 41.7 | 20.2 |
| M2m-ERM (Kim et al., 2020) | 42.9 | - | 58.2 | - | - | - |
| M2m-LDAM (Kim et al., 2020) | 43.5 | - | 57.6 | - | - | - |
| cRT (Kang et al., 2019) | 43.3 | 46.8 | 58.1 | 64.0 | 44.8 | 18.1 |
| LWS (Kang et al., 2019) | 43.1 | 46.4 | 58.1 | - | - | - |
| SCL (Khosla et al., 2020) † | 42.1 | 45.2 | 54.8 | 62.8 | 42.0 | 18.4 |
| KCL (Kang et al., 2020) | 42.8 | 46.3 | 57.6 | - | - | - |
| KCL† | 42.8 | 46.4 | 57.5 | 63.4 | 42.5 | 19.2 |
| TSC (Li et al., 2022) | 43.8 | 47.4 | **59.0** | - | - | - |
| TSC† | 43.5 | 47.6 | 58.7 | 63.7 | 43.2 | 20.4 |
| SBCL | **44.9**±0.3 | **48.7**±0.2 | 57.9±0.2 | **64.4**±0.3 | **45.3**±0.2 | **22.2**±0.3 |

learning. Table 2 summarizes the results on CIFAR-100-LT dataset. For CIFAR-100-LT dataset, SBCL outperforms previous SOTA methods except for imbalance ratio 10. We hypothesize that it is because the tail class of CIFAR-100-LT with imbalance ratio 10 has multiple samples, which makes it hard to distinguish the performance of methods on the long-tailed recognition.

### 4.3 PERFORMANCE ON OTHER VISUAL TASKS

There is a recent trend of using the contrastive learning to pretrain a feature extractor for downstream visual tasks other than image classification (He et al., 2020). We are curious about two questions: (1) when the pretraining dataset is class-imbalanced, how the downstream performance is affected? (2) In such case, can our SBCL improve the learned feature extractor over existing contrastive learning baselines? To answer these questions, we use the object detection task of PASCAL VOC dataset as the evaluation suite and use ImageNet/ImageNet-LT datasets as class-balanced/-imbalanced pretraining datasets. Following Kang et al. (2020); He et al. (2020), we first pretrain a feature extractor on ImageNet/ImageNet-LT then further finetune it for the downstream object detaction tasks using Faster R-CNN (Ren et al., 2015) with R50-C4 backbone.

The experiment results are shown in Tables 3. From the results, we can see that pretraining on class-balanced data (ImgeNet) leads to consistently better results than that on class-imbalanced dataset (ImageNet-LT) pretraining the model on the ImageNet and ImageNet-LT datasets by the SBCL can perform slightly better than other baselines. In addition, the proposed SBCL significantly outperforms baselines on class-imbalanced pretraining dataset, while achieve comparable performance on class-balanced ones. For the representation which trained on the full ImageNet dataset, the performance advantage is not obvious. In Appendix A.1, we show additional experimental results of object detection and instance segmentation on COCO (Lin et al., 2014) dataset and SBCL also outperforms other baseline methods. Thus, we conclude that the proposed SBCL is not only helpful for image classification, but also other visual tasks.

### 4.4 ANALYSIS OF FEATURE DISTRIBUTION

To analyze the representation learned by SBCL, we firstly define the euclidean distance between a given sample and other samples from the same/different classes as intra/inter-class distance. Concretely, the euclidean distance between a sample $z_i$ and a set $S$ is defined as $\mathbf{D}(z_i, S) = \frac{1}{|S|} \sum_{z_j \in S} \|z_i - z_j\|_2$. Then, the intra- and inter-class distance of sample $z_i$ can be defined as

$\mathbf{D}(z_i, P_i)$ and $\mathbf{D}(z_i, \mathcal{D}/P_i)$ separately; and the intra- and inter-subclass distance of sample $z_i$ can be defined as $\mathbf{D}(z_i, M_i)$ and $\mathbf{D}(z_i, P_i/M_i)$ separately.

Table 3: Object detection Results on PASCAL VOC.

| Method | ImageNet | | | ImageNet-LT | | |
|---|---|---|---|---|---|---|
| | AP$_{50}$ | AP | AP$_{75}$ | AP$_{50}$ | AP | AP$_{75}$ |
| random init. | 60.2 | 33.8 | 33.1 | 60.2 | 33.8 | 33.1 |
| CE | 81.3 | 53.7 | 59.2 | 76.5 | 48.5 | 51.0 |
| CL (He et al., 2020) | 81.3 | 56.1 | 62.7 | 78.2 | 51.5 | 56.5 |
| KCL (Kang et al., 2020) | 82.3 | 55.5 | 62.1 | 79.7 | 52.6 | 57.9 |
| SBCL | 81.9±0.2 | 56.2±0.2 | 62.8±0.1 | 80.6±0.2 | 53.4±0.2 | 58.8±0.1 |

Table 4: Ablation study on different components of SBCL.

| Warm-up | Adaptive cluster | Dynamic temperature | CIFAR-100-LT | | |
|---|---|---|---|---|---|
| | | | Imbalance Ratio | | |
| | | | 100 | 50 | 10 |
| | ✓ | ✓ | 44.0±0.3 | 47.9±0.3 | 57.2±0.3 |
| ✓ | | ✓ | 43.8±0.2 | 47.2±0.3 | 56.5±0.2 |
| ✓ | ✓ | | 43.7±0.2 | 47.8±0.2 | 57.0±0.2 |
| ✓ | ✓ | ✓ | 44.9±0.3 | 48.7±0.2 | 57.9±0.2 |

Table 5: Average intra-class/inter-class distance of features learned with different contrastive learning methods.

| Distance | Method | Many | Medium | Few | All |
|---|---|---|---|---|---|
| | KCL | 0.44 | 0.53 | 0.70 | 0.46 |
| Intra-class | TSC | 0.60 | 0.62 | 0.77 | 0.61 |
| | SBCL | **1.00**±0.01 | **0.94**±0.02 | **0.88**±0.02 | **0.99**±0.01 |
| | KCL | 1.27 | 1.26 | 1.24 | 1.26 |
| Inter-class | TSC | 1.32 | 1.31 | 1.29 | 1.32 |
| | SBCL | **1.39**±0.01 | **1.38**±0.01 | **1.37**±0.01 | **1.39**±0.01 |

**Intra-class/Inter-class distance.** SBCL aims at learning a compact representation space, in which representations from different classes are far from each other and the feature space spanned by representations of each class is invariant to the long-tailed distribution. Though the intra/inter-class distance on the feature space, we compare SBCL on the previous methods (KCL ,TSC) on CIFAR-100-LT with imbalance ratio 100. The results of the average distance are summarized in Table 5 and the distances of different groups are reported separately. The results show several good properties of SBCL over previous methods: (i) the inter-class distances of SBCL is larger than the previous methods, which implies that SBCL can push different classes far away from each other, and thus help the downstream tasks; (ii) the intra-class distance of SBCL is relatively more equal in different disjoint subsets than previous methods, which indicates that that head/tail classes have similar volume of the learned space and thus help balance different classes.

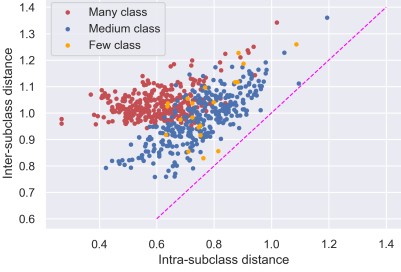

(a) Feature distance in subclasses.

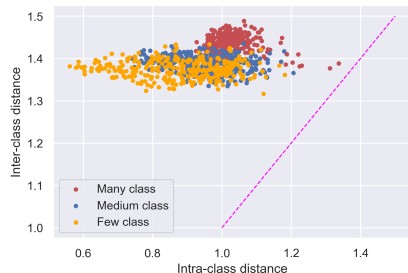

(b) Feature distance in classes.

Figure 2: Feature distance of subclasses and classes on CIFAR-100-LT with imbalance ratio 100. We randomly sample instances from many-shot and medium-shot classes so that the size of each equals to that of few-shot classes.

**Feature distribution of SBCL.** As shown in Figure 2a, the distance between samples from the same subclass is less than those from the same class but different subclasses. Meanwhile, in Figure 2b, the inter-class distance is higher than the intra-class distance with stable value, which denotes features from the different classes are uniformly distributed on a hypersphere. The results in all indicate that the two-layer class hierarchy are successfully captured and feature distribution achieves the core idea of SBCL.

## 4.5 ABLATION STUDIES

**Warm-up.** As mentioned in Section 3.3, we train the feature extractor for several epochs using ordinary SCL or KCL as warm-up stage. As shown in Table 4, such a warm-up stage is beneficial since the performance drops when we remove the warm-up stage. This is likely because at the early stage of training, the extracted feature is not well-trained and the cluster assignment could be noisy and ineffective, hindering the efficacy of SBCL.

**Adaptive clustering.** We are also curious about the efficacy of adaptive clustering and thus present the performance of SBCL with clustering only once and fixing cluster assignments during training. As shown in Table 4, without adaptive clustering, the performance decreases in all cases. The reason could be fixed cluster assignment is prone to noise when the model is not well-trained, while adaptive clustering would dynamically adjust the cluster assignments based on the current learned model, which is supposed to become better as the training proceeds.

**Dynamic temperature.** In Table 4, we also study the effectiveness of dynamic temperature (Section 3.2). We remove the dynamic temperature and simply set $\tau_2 > \tau_1$ following (Hoffmann et al., 2022). With the fixed temperature $\tau_2$, the performance of SBCL is significantly worse than that with dynamic temperature. We speculate that this is because dynamic temperature could help prevent the instances of a class to form overly large or small cluster in the feature space and therefore lead to better learned representations. Additionally, to evaluate the impact of dynamic temperature on other baselines, we apply the dynamic temperature on TSC, as reported in Appendix A.1.

## 5 RELATED WORK

Traditional methods for handling long-tailed recognition problem includes re-sampling and re-weighting. There are roughly two types of re-sampling techniques: over-sampling the minority classes (Shen et al., 2016; Zhong et al., 2016; Buda et al., 2018; Byrd & Lipton, 2019) and under-sampling the frequent classes (He & Garcia, 2009; Japkowicz & Stephen, 2002; Buda et al., 2018). The re-weighting techniques assign adaptive weights for different classes or even different samples. The vanilla scheme re-weights classes proportionally to the inverse of their frequency (Huang et al., 2016; 2019; Wang et al., 2017). For class-level re-weighting methods, many loss functions including CB loss (Cui et al., 2019), LDAM loss (Cao et al., 2019) and Balanced softmax loss (Ren et al., 2020) were recently proposed, while instance-level re-weighting methods include Focal loss (Lin et al., 2017) and Influence-balanced loss (Park et al., 2021). Recently, two-stage algorithms have achieved remarkable performance for long-tailed recognition, such as classifier re-training (cRT) (Kang et al., 2019), learnable weight scaling (LWS) (Kang et al., 2019), and Mixup Shifted Label-Aware Smoothing model (MiSLAS) (Zhong et al., 2021). Meanwhile, bilateral branch network (BBN) (Zhou et al., 2020) uses an additional network branch for re-balancing. RIDE (Wang et al., 2021a) use multiple branches named experts, each learning to specialize in the entire classes. LADE (Hong et al., 2021) assumes the prior of test class distributions is available and accordingly post-adjust model predictions. PaCo (Cui et al., 2021) applies parametric class-wise learnable centers to rebalance in contrastive learning. BCL (Zhu et al., 2022) proposes a multi-branch framework to achieve class-averaging and class-complement in the training process.

To boost the performance of the two-stage algorithms, researchers have introduced supervised contrastive learning (Khosla et al., 2020) to the first feature-learning stage and proposed $k$-positive contrastive loss (KCL) (Kang et al., 2020) and targeted supervised contrastive learning (TSC) (Li et al., 2022). While achieving the state-of-the-art performance, these methods inject class-balance in the contrastive learning objective, inevitably leading to instance-imbalance during training. In this work, we instead propose to achieve both subclass- and instance-balance in the contrastive learning object. Our method is also related to recent studies of clustering-based deep unsupervised learning (Dosovitskiy et al., 2014; Xie et al., 2016; Liao et al., 2016; Yang et al., 2016; Caron et al., 2018; 2020), especially those that leverage contrastive learning (Li et al., 2021b;a; Wang et al., 2021b; Guo et al., 2022). However, they target at general unsupervised representation learning scenario, while our method is tailored for long-tailed recognition where the training data is immensely class-imbalanced.

## 6 CONCLUSION

In this paper, we introduced Subclass-balancing Contrastive Learning (SBCL) for long-tailed recognition. It breaks down the head classes into multiple semantically-coherent subclasses via subclass-balancing adaptive clustering and incorporates a bi-granularity contrastive loss that encourages both subclass- and instance-balance. Extensive experiments on multiple datasets demonstrate that SBCL achieves state-of-the-art single-model performance on benchmark datasets for long-tailed recognition.

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

# A APPENDIX

## A.1 ADDITIONAL EXPERIMENT RESULTS

**Accuracy of each class on CIFAR-100-LT.** We visualize the accuracy of each class of both SCL and SBCL on CIFAR-100-LT with imbalance ratio 100 (Figure 3). From the results, we can see that SBCL improves performance on tail classes over SCL without the expense of the perforamnce of the head classes.

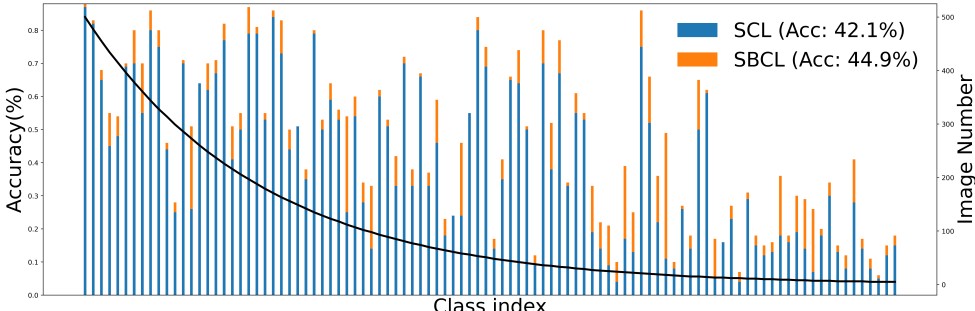

Figure 3: Accuracy of each classes on CIFAR100-LT. The black line is the class distribution, and the classes in the left part are head classes while those in the right part are tail classes.

**The per-class weight norm of a linear classifier trained on top of features learned by SBCL in different training stages.** Figure 4 shows the per-class weight norm of a linear classifier trained on top of features learned by SBCL in different training stages on CIFAR-100-LT. From the figure, we can see that as the training proceeds, the per-class weight norm becomes model balanced even when training the linear classifier, the original cross-entropy loss is used.

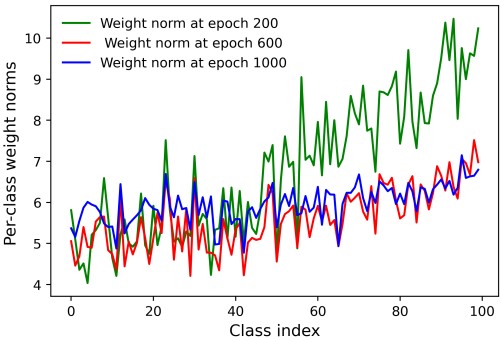

Figure 4: Change in weight norm of the linear classifier based on the representations trained by SBCL on CIFAR-100-LT with imbalance ratio 100 during training.

Table 6: Different selection of negative instances in SBCL on CIFAR-100-LT with different imbalance ratio. The 'Class label' row are the first term of loss function is constructed by class label and the 'Subclass label' row are subclass label.

| Negative samples | Imbalance Ratio | | |
|---|---|---|---|
| | 100 | 50 | 10 |
| Class label | 43.7 | 47.6 | 56.8 |
| Subclass label | **44.9** | **48.7** | **57.9** |

**Selection of negative instances in SBCL.** Our proposed loss in Eq. 4 consists of two supervised contrastive losses with subclass and class labels respectively. The first term regards instances in

different subclasses as negative instead of these in different classes; In Table 6 we show that such design choice leads to better performance than using instances in different classes as negative, which illustrates the effectiveness of exploiting the rich semantic in head classses.

**Intra-subclass/Inter-subclass distance.** To leverage instance semantic coherence to balance the feature space, we expect instances of high semantic coherence to form a more concentrated cluster than other instances in the same class. So, we embed the subclass-balancing adaptive clustering strategy on previous methods to illustrate this on CIFAR-100-LT with imbalance ratio 100. In Table 7, we report the intra-subclass/inter-subclass distance on different splits. Compare with KCL and TSC, the results show that SBCL achieves to concentrate instances from the same subclass and pulls instances from different subclasses away on all splits. We also note that the inter-subclass distance of SBCL is invariant with the decreasing of group split. This means the head class could be split into many subclasses separately, which constructs a balance feature space for the long-tailed recognition.

Table 7: Intra-subclass/inter-subclass distance of features learned with different contrastive learning methods.

| Metric | Method | Many | Medium | Few | All |
|---|---|---|---|---|---|
| Intra-subclass distance | KCL | 0.27 | 0.40 | 0.69 | 0.30 |
| | TSC | 0.40 | 0.48 | 0.76 | 0.42 |
| | SBCL | **0.68**±0.02 | **0.76**±0.02 | **0.87**±0.03 | **0.70**±0.02 |
| Inter-subclass distance | KCL | 0.46 | 0.59 | 0.70 | 0.48 |
| | TSC | 0.61 | 0.69 | 0.77 | 0.63 |
| | SBCL | **1.02**±0.01 | **0.99**±0.01 | **0.89**±0.02 | **1.01**±0.01 |

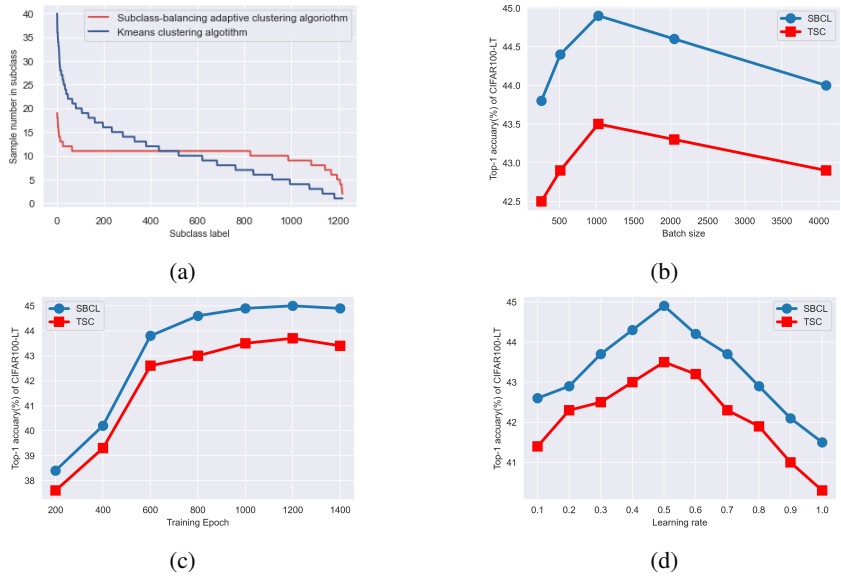

(a)

(b)

(c)

(d)

Figure 5: **Analysis of SBCL as a loss function of different hyperparameters on CIFAR-100-LT with imbalance ratio 100.** (a): Sample number in clusters with different cluster algorithm. (b): Top-1 accuracy of SBCL/TSC as a function of different batch size. (c): Top-1 accuracy of SBCL/TSC as a function of different pretraining epochs. (d): Top-1 accuracy of SBCL/TSC as a function of different learning rates.

**Hyperparameter analysis on CIFAR-100-LT.** Figure 5a and Table 8 show the distribute situation of sample number in subclasses obtained by different cluster algorithms on CIFAR-100-LT with imbalance ratio 100. For Kmean cluster algorithm, the imbalance phenomenon of subclasses is obvious. When using our proposed cluster algorithm, the imbalance ratio of sample number in subclasses deceases from 40 to 9.5. And the standard deviation of sample number on CIFAR-100-LT

is relatively small, which denotes the number of samples in most subclasses keeps stable in a certain range.

Figure 5b shows the impact of batch size on SBCL/TSC. We find that larger batch sizes have a significant advantage over the smaller ones. This is because larger batch sizes provide more negative examples to facilitate convergence. However, the over-large batch size hurts the model performance. And SBCL and TSC are equally sensitive to batch size on CIFAR-100-LT.

Figure 5c shows the curve of the accuracy of SBCL/TSC vs. the number of training epochs. From the curve, we can see that the performance of SBCL and TSC both converge after 800 epochs. When the model is trained with SBCL over 600 epochs, its performance already exceeds TSC.

In Figure 5d, we display the performance of SBCL with different learning rates on CIFAR-100-LT with imbalance ratio 100. As shown in the figure, the learning rate has significant impact on the performance, and we set the learning rate as 0.5 for CIFAR-100-LT.

Table 8: Distribution of sample number in subclass on CIFAR-100-LT with imbalance ratio 100.

| Dataset | Max | Min | Average | Std | Imbalance ratio(Max/Min) |
|---|---|---|---|---|---|
| Kmean clustering algorithm | 40 | 1 | 10.34 | 6.27 | 40 |
| Subclass-balancing adaptive clustering algorithm | 19 | 2 | 10.34 | 1.60 | 9.5 |

**Combining TSC with dynamic temperature.** According to Table 4, dynamic temperature effectively contributes to the improvement of accuracy. We also add the dynamic temperature to the second term of TSC (Li et al., 2022) and the experiment results are shown in Table 9. However, the improvement of the dynamic temperature on TSC is less significant than that on our method, which is reasonable because we introduce dynamic temperature for the loss to distinguish between class and subclass, while TSC does not have subclass and therefore the dynamic temperature is less effective.

Table 9: Combination of TSC and SBCL with dynamic temperature.

| Dynamic temperature | CIFAR-100-LT | | | | | |
|---|---|---|---|---|---|---|
| | TSC | | | SBCL | | |
| Imbalance Ratio | 100 | 50 | 10 | 100 | 50 | 10 |
| | 43.5 | 47.6 | 58.7 | 43.7 | 47.8 | 57.0 |
| ✓ | 43.9(+0.4) | 48.0(+0.4) | 59.2(+0.5) | 44.9(+1.2) | 48.7(+0.9) | 57.9(+0.9) |

**Warm-up on ImageNet-LT.** Instead of using the SCL at the warm-up stage for CIFAR datasets, KCL is adopted for ImageNet-LT and iNaturalist 2018 datasets to warm up the feature extractor. As Table 10 shows, warm-up phase makes feature extractor improve accuracy on all splits of ImageNet-LT. This is because it prevents cluster assignment from feature random distribution at the begining and avoids using the SCL to make the feature space dominated by the head class at the warm-up stage.

Table 10: SBCL with and without warm-up stage on ImageNet-LT.

| Method | Many | Medium | Few | All |
|---|---|---|---|---|
| SBCL without warm-up | 62.9 | 49.6 | 29.3 | 52.0 |
| SBCL with warm-up | **63.8** | **51.3** | **31.2** | **53.4** |

**Advantages of cluster validity.** Actually, previous studies (Kang et al., 2020; Li et al., 2022) have proven that randomly sampling balanced instances as positive pairs (such as KCL, TSC) is better than sampling all instances of the same class as positive pairs (such as SCL). However, this strategy may destruct instance semantic coherence. In Table 11, we replace the first team (regard subclasses as positive pairs) with the balanced positive sampling strategy (KCL) to prove this on ImageNet-LT. As the results show, subclass-balancing adaptive clustering strategy brings more improvement to SBCL than balanced positive sampling strategy.

Table 11: Subclass-balancing adaptive clustering strategy improves more than balanced positive sampling strategy on ImageNet-LT.

| Method | Many | Medium | Few | All |
|---|---|---|---|---|
| FCL | 61.4 | 47.0 | 28.2 | 49.8 |
| KCL | 62.4 | 49.0 | 29.5 | 51.5 |
| TSC | 63.5 | 49.7 | 30.4 | 52.4 |
| SBCL (KCL) | 63.3 | 49.5 | 30.6 | 52.2 |
| SBCL | **63.8** | **51.3** | **31.2** | **53.4** |

**COCO object detection and instance segmentation.** In this section, following the experiment setting in (He et al., 2020), we use Mask R-CNN (He et al., 2017) to conduct the object detection and instance segmentation experiments on COCO dataset. The schedule is the default 2× in (He et al., 2020). Table 12 shows the pretrained model trained by SBCL outperforms it learned with other contrastive learning for the downstream tasks.

Table 12: **Object detection and instance segmentation results on COCO dataset.** The representation model is trained on ImageNet and ImageNet-LT. We report results in bounding-box AP ($AP^{bb}$) and mask AP ($AP^{mk}$).

| | Method | ImageNet | | | ImageNet-LT | | |
|---|---|---|---|---|---|---|---|
| | | AP | $AP_{50}$ | $AP_{75}$ | AP | $AP_{50}$ | $AP_{75}$ |
| $AP^{bb}$ | random init. | 35.6 | 54.6 | 38.2 | 35.6 | 54.6 | 38.2 |
| | CE | 40.1 | 59.8 | 43.3 | 38.1 | 57.4 | 41.2 |
| | CL (He et al., 2020) | 40.4 | 60.1 | 44.1 | 39.7 | 59.4 | 42.7 |
| | KCL (Kang et al., 2020) | 40.8 | 60.6 | 44.0 | 39.4 | 59.1 | 42.6 |
| | SBCL | **41.1**±0.2 | **60.8**±0.2 | **44.2**±0.1 | **40.0**±0.2 | **59.6**±0.2 | **43.0**±0.1 |
| $AP^{mk}$ | random init. | 31.4 | 51.5 | 33.5 | 31.4 | 51.5 | 33.5 |
| | CE | 34.9 | 56.6 | 37.0 | 33.3 | 54.2 | 35.4 |
| | CL (He et al., 2020) | 35.1 | 56.9 | 37.6 | 34.7 | 56.1 | 37.1 |
| | KCL (Kang et al., 2020) | 35.5 | 57.4 | 37.8 | 34.4 | 55.8 | 36.4 |
| | SBCL | **35.7**±0.2 | **57.5**±0.2 | **37.9**±0.1 | **35.0**±0.2 | **56.3**±0.1 | **37.3**±0.1 |

**Combining SBCL with ensemble-based methods.** Another line of research to address the long-tailed problem is the ensemble-based methods, such as RIDE (Wang et al., 2021a), which incorporates multiple models in a multi-expert framework. Here we show that SBCL can also be leveraged to boost the performance of RIDE (Wang et al., 2021a), a state-of-the-art ensemble-based method. To implement SBCL with RIDE, we follow (Li et al., 2022) to simply replace the feature extractor on stage-1 training in RIDE with that trained with SBCL and keep the stage-2 routing training unchange. As shown in Table 13, applying SBCL to RIDE improves its performance with a significant gap, outperforms the combination of TSC and RIDE on all different number of experts.

Table 13: Performance of the combination of SBCL and state-of-the-art ensemble-based method RIDE (Wang et al., 2021a) with ResNet-50 (He et al., 2016) on ImageNet-LT.

| Method | Many | Medium | Few | All |
|---|---|---|---|---|
| RIDE (2 experts) | 65.8 | 51.0 | 34.6 | 54.4 |
| RIDE (3 experts) | 66.2 | 51.7 | 34.9 | 54.9 |
| RIDE (4 experts) | 66.2 | 52.3 | 36.5 | 55.4 |
| TSC+ RIDE (2 experts) | 68.4 | 51.3 | 36.4 | 55.9 |
| TSC+ RIDE (3 experts) | 69.1 | 51.7 | 36.7 | 56.3 |
| TSC+ RIDE (4 experts) | 69.2 | 52.4 | 37.9 | 56.9 |
| SBCL + RIDE (2 experts) | 68.6 | 51.9 | 36.5 | 56.2 |
| SBCL + RIDE (3 experts) | 69.2 | 52.4 | 36.9 | 56.8 |
| SBCL + RIDE (4 experts) | **69.5** | **52.6** | **38.1** | **57.1** |

**Combining SBCL with PaCo (Cui et al., 2021) and BCL (Zhu et al., 2022).** PaCo (Cui et al., 2021) and BCL (Zhu et al., 2022) proposed new variants of supervised contrastive loss and jointly train both the proposed loss and classification loss to improve long-tail recognition, while we focus on the two stage pipeline, especially the first stage of representation learning. In this experiment,

we show that using models pretrained with both TSC and SBCL as initialization could improve the performance of both PaCo and BCL. The results can be found in Table 14, and we can see that SBCL renders larger performance gain than TSC. The improvement over PaCo and BCL demonstrates the effectiveness of SBCL in long-tail recognition, and shed lights on potential future work to evaluate the combination of multiple techniques of long-tail recognition to achieve new SOTA results.

Both PaCo and BCL adopt a single-stage pipeline where a classifier is trained with both classification loss and supervised contrastive loss. To further test the effectiveness of SBCL, we replace the supervised contrastive loss they used with SBCL and retrain all other techniques they proposed for fair comparison. The results can be found in Table 15. We can see that using SBCL could improve over PaCo and BCL.

Table 14: Performance of the combination of SBCL and extended contrastive methods (PaCo (Cui et al., 2021) and BCL (Zhu et al., 2022))) with ResNext-50 (Saining et al., 2017) on ImageNet-LT.

| Method | Many | Medium | Few | All |
|---|---|---|---|---|
| PaCo | 64.4 | 55.7 | 33.7 | 56.0 |
| BCL | 67.9 | 54.2 | 36.6 | 57.1 |
| TSC+ PaCo | 66.4 | 55.8 | 35.7 | 57.1 |
| TSC+ BCL | 69.0 | 56.3 | 37.8 | 58.7 |
| SBCL + PaCo | 66.9 | 56.1 | 38.4 | 57.9 |
| SBCL + BCL | **69.5** | **56.7** | **39.0** | **59.2** |

Table 15: Top-1 accuracy on CIFAR-100-LT with ResNet-32 (He et al., 2016).

| Method | Many | Medium | Few | All |
|---|---|---|---|---|
| CE | 65.2 | 37.1 | 9.1 | 38.3 |
| PaCo | - | - | - | 52.0 |
| BCL | 69.7 | 53.8 | 35.5 | 53.9 |
| SBCL + PaCo | 68.3 | 54.8 | 30.8 | 52.3 |
| SBCL + BCL | **69.4** | **54.5** | **37.6** | **54.5** |

**Combining SBCL with two-stage methods.** MisLAS (Zhong et al., 2021) provides two major technical contributions to improve the two-stage pipeline of the longtail recognition, *i.e.*, label-aware smoothing and shifted batch normalization. Both techniques are designed for improving the second stage (the classifier learning), while our method is to improve the first stage of representation learning. Thus, MisLAS could be combined with SBCL and TSC. In Table 16, we report the results of combing MisLAS with both SBCL and TSC. The results show that both SBCL and TSC improve the performance of MisLAS and SBCL renders more performance boost than TSC.

Table 16: Performance of the combination of SBCL and MisLAS (Zhong et al., 2021) with ResNet-50 (He et al., 2016) on ImageNet-LT.

| Method | Many | Medium | Few | All |
|---|---|---|---|---|
| MiSLAS | - | - | - | 52.7 |
| TSC+ MiSLAS | 63.7 | 50.5 | 36.0 | 53.6 |
| SBCL + MiSLAS | **64.1** | **52.0** | **36.4** | **54.5** |

**Freezing the pretrained model for object detection.** We assess the representation trained on ImagNet/ImageNet-LT for the downstream detection task. We freeze the pretrained backbone to train the Faster R-CNN (Ren et al., 2015), and use the same training schedule in Section 4.3. Table 17 reports the average mAP on PASCAL VOC dataset. The representation trained by SBCL obtains better performance for object detection.

**Hyperparameter studies.** Here, we study the effect of hyperparameters $\beta$ and $\delta$. Note that $\beta$ controls the balance of two loss terms in Eq. 4 and $\delta$ determines the lower bound of the cluster size in Eq. 3. Specifically, on CIFAR-100-LT with imbalance ratio 100, we vary the values of $\beta$ from {0.1,

Table 17: Object detection on PASCAL VOC when the pretrained model is frozen.

| Method | Pretrain dataset | mAP |
|---|---|---|
| Supervised | ImageNet | 79.3 |
| | ImageNet-LT | 70.9 |
| KCL (Kang et al., 2020) | ImageNet | 80.5 |
| | ImageNet-LT | 73.4 |
| SBCL(ours) | ImageNet | **80.7** |
| | ImageNet-LT | **74.5** |

0.2, 0.5, 0.8, 1.0, 2.0} with $\delta = 10$ and the value of $\delta$ from {5, 10, 20, 30, 50, 100} with $\beta = 0.2$. The results are summarized in Table 18. We observe that the smaller $\beta$ values (between 0.1 and 0.5) can achieve relatively good performance, with the best being 0.2. This observation aligns with our intuition of emphasizing the subclass-level contrastive loss, because smaller $\beta$ is equivalent to putting more weights on the first term of Eq. 4, which corresponds to the subclass-level contrastive. For $\delta$, the values between 5 and 30 yield high accuracy, with the best being 10. We can see that large $\delta$ values ($\delta = 50, 100$) lead to significant drop in performance. We argue that this is because large $\delta$ value would result in subclasses that contain more instance than tail classes and therefore affect the subclass-balance, leading to suboptimal performance. In addition, smaller $\delta$ value ($\delta = 5$) also causes performance drop; the reason could be small cluster size may let similar instance being assigned to different clusters and therefore affect the learned representations. Therefore, we fix $\beta = 0.2$ and $\delta = 10$ for all experiments.

Table 18: Hyperparameter study of $\beta$ and $\delta$ on CIFAR-100-LT with imbalance ratio 100.

| $\beta$ | 0.1 | 0.2 | 0.5 | 0.8 | 1.0 | 2.0 |
|---|---|---|---|---|---|---|
| ACC(%) | 44.6 | **44.9** | 44.5 | 44.1 | 43.9 | 42.1 |
| $\delta$ | 5 | 10 | 20 | 30 | 50 | 100 |
| ACC(%) | 44.3 | **44.9** | 44.6 | 44.3 | 42.9 | 42.3 |

**Visualization of generated clusters.** In Figure 6, we show the clustering results of ImageNet-LT training images generated by subclass-balancing adaptive clustering algorithm. From the results, we can see that the algorithm is able to find the subclasses with similar patterns, helping the model learn semantic coherent representations. For example, the two subclasses in the bottom-left are telephone with/without human.

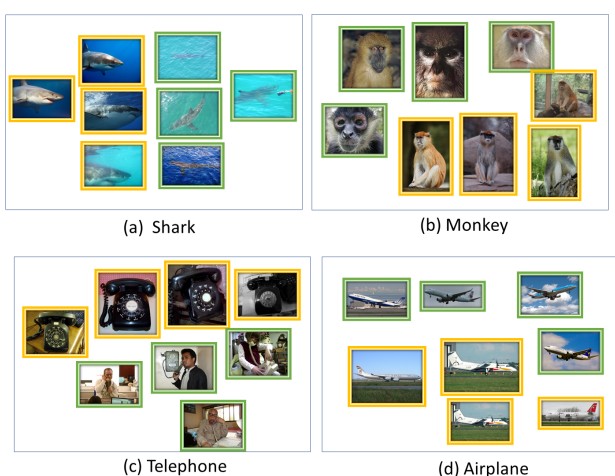

(a) Shark

(b) Monkey

(c) Telephone

(d) Airplane

Figure 6: Visualization of subclasses generated by SBCL. Images with green and orange boarder are randomly drawn from different subclasses within the same classes. We can see that SBCL could produce semantically coherent subclasses.

## A.2 ADDITIONAL INFORMATION

**Benchmark datasets statistical information and Implementation Details.** We summarize the statistical information of the three benchmark datasets in Table 19. Following (Kang et al., 2019; 2020; Li et al., 2022), we apply SBCL on the long-tailed recognition by using a two-stage training strategy: (i) train the representation with SBCL; (ii) learn a linear classifier on top of the fixed representation. The training process is the same as TSC (Li et al., 2022). Thus, we use TSC default hyperparameters and implementation details for the representation learning. For CIFAR-100-LT dataset, all experiments are performed on 2 NVIDIA RTX 3090 GPUs. For ImageNet-LT and iNaturalist 2018 datasets, we perform the experiments on 8 NVIDIA RTX 3090 GPUs. The detailed hyperparameters of TSC and SBCL are given in Table 20.

For the classify learning, training the linear classifier strategy is the same with TSC (Li et al., 2022); so, we use TSC default hyperparameters and implementation details for the classifier learning. For the detect model learning, we follow MoCo (He et al., 2020) to adopt the same setting, hyperparameters and evolution metrics with R50-C4 backbone. For Pascal VOC dataset, we train Faster R-CNN (Ren et al., 2015) on VOC07+12 and evaluate on the test set of VOC07. For COCO dataset, we train Mask R-CNN (He et al., 2017) on train2017 set and evaluate on val2017 set.

Table 19: Statistics of datasets. The imbalance ratio $\rho = n_1/n_C$.

| Dataset | classes | training data | test data | imbalance ratio |
|---|---|---|---|---|
| CIFAR-100-LT | 100 | 50,000 | 10,000 | $\{100, 50, 10\}$ |
| ImageNet-LT | 1,000 | 115,846 | 50,000 | 256 |
| iNaturalist 2018 | 8,142 | 437,513 | 24,426 | 500 |

Table 20: Hyperparameters used by different loss functions for benchmark datasets. The detailed hyperparameters of iNaturalist 2018 are the same as the ImageNet-LT.

| Hyperparameters | ImageNet-LT | | CIFAR100-LT | |
|---|---|---|---|---|
| | TSC | SBCL | TSC | SBCL |
| module | MoCo | MoCo | SimCLR | SimCLR |
| warm-up epoch | 200 | 200 | 0 | 10 |
| epoch | 400 | 400 | 1000 | 1000 |
| batch size | 256 | 256 | 1024 | 1024 |
| learning rate | 0.1 | 0.1 | 0.5 | 0.5 |
| learning rate schedule | cosine | cosine | cosine | cosine |
| memory size | 65536 | 65536 | - | - |
| encoder momentum | 0.999 | 0.999 | - | - |
| feature dimension | 128 | 128 | 128 | 128 |
| softmax temperature | 0.07 | 0.07 | 0.1 | 0.1 |
| $k$-positive number | 6 | - | 4 | - |
| hyperparameter of $\beta$ | 0.2 | 0.2 | 0.2 | 0.2 |
| hyperparameter of $\delta$ | - | 20 | - | 10 |
| hyperparameter of $\alpha$ | - | 10 | - | 10 |

**Limitations.** SBCL has some limitations. First, clustering the head class in SBCL takes a long time on the training phase, especially for ImageNet-LT and iNaturalist 2018. Second, SBCL requires knowing the number of samples in each class to decide the cluster number; so, it is not applicable to problems where the number of samples is unknown.

**Social impacts.** This work aims to propose a novel representation learning to help people resolve the bias in the real world data recognition, which might has positive social impact. We do not foresee any form of negative social impact induced by our work.

**Privacy information in data.** All datesets we used in the experiment are public. The datasets only include the pictures, which most are animals and plants. No private information is included.

**Baseline information.** We report the accuracy of KCL and TSC on different benchmark datasets from (Li et al., 2022). For SwAV[1] (Caron et al., 2020), PCL[2] (Li et al., 2021a) and BYOL[3] (Grill et al., 2020), we use their official open-source implementations.

---

[1]SwAV offical implementation: https://github.com/facebookresearch/swav.

[2]PCL offical implementation: https://github.com/salesforce/PCL.

[3]BYOL offical implementation: https://github.com/deepmind/deepmind-research/tree/master/byol.

