# OpenReview forum: "Subclass-balancing Contrastive Learning for Long-tailed Recognition"
_ICLR.cc/2023/Conference — Submitted to ICLR 2023_

### Official Review · Reviewer_aZoT · 2022-10-21

**Confidence:** 4
**Correctness:** 4
**Technical Novelty And Significance:** 3
**Empirical Novelty And Significance:** 2
**Recommendation:** 6

**Clarity, Quality, Novelty And Reproducibility:**

This paper is clearly presented and the somewhat novel.
The method seems easy to reproduce.

Additional concerns:
(1) What if using adaptive clustering for cross-entropy based long-tailed recognition?

**Strength And Weaknesses:**

Strength:

(1) The motivation of this work, i.e., dividing head classes to sub-classes to deal with class imbalance, is clear. The basic idea of enforcing both instance- and subclass-balance by leveraging supervised contrastive learning is reasonable.

(2) The paper is well organized and clearly presented. The method seems easy to reproduce as well.

(3) Experiments on LT datasets and downstream tasks are adequate.

Weaknesses:

(1) Though the motivation and proposed method are reasonable, the comparison with similar long-tailed contrastive learning approaches is not complete. In addition to KCL and TSC, there are some other related solutions for long-tailed recognition problem, like PaCo [1] and BCL [2]. What is the advantage of sub-class clustering over these solutions?

(2) There are quite a few hyper-parameters, like temperature, $\beta$, $\delta$ and $\alpha$ in Eq. 5. Though the authors provide some hyper-parameter study on CIFAR-100-LT, I still wonder how the hyper-parameters change across datasets? Do I need to re-adjust these hyper-parameters with lots of efforts for a new dataset to achieve good performance?

(3) In Table 1, it seems these contrastive methods are implemented on different code base. To verify the effectiveness of SBCL over the SCL baseline, it is better to use your own implementation on SCL.

(4) There are quite a few works trying to re-balance the supervised contrastive loss for long-tailed recognition. What if we instead use non-contrastive methods like BYOL? For non-contrastive methods, the subclass-balancing problem naturally disappears.


[1] Parametric contrastive learning, ICCV 2021.

[2] Balanced Contrastive Learning for Long-Tailed Visual Recognition, CVPR 2022.


**Summary Of The Paper:**

This paper proposes subclass-balancing contrastive learning (SBCL) for long-tailed recognition. It divides head classes into several fine-grained subclasses via adaptive clustering, to enforce equal probability of head/tail classes to be engaged. After clustering, a bi-granularity contrastive loss is proposed. In addition, dynamic temperature and warm-up are also used for SBCL. Experimental results show the effectiveness of the proposed method.

**Summary Of The Review:**

The motivation is clear. The proposed method is reasonable though some related works have similar ideas.
There are some weak aspects in empirical study, as pointed out above.

---

> ### Author Response · Authors · 2022-11-15
> **Response to Reviewer aZoT**
>
> **Though the motivation and proposed method are reasonable, the comparison with similar long-tailed contrastive learning approaches is not complete. In addition to KCL and TSC, there are some other related solutions for long-tailed recognition problem, like PaCo [1] and BCL [2]. What is the advantage of sub-class clustering over these solutions??**
>
> Thanks for proposing the baseline to be compared.
> We have added these baselines in Table 14 though combining SBCL with these extended contrastive methods.
> From the experiment results, SBCL could improve the baseline performance with a significant gap.
> And we also replace the supervised contrastive learning in PaCo and BCL with SBCL loss and Table 15 shows SBCL improves over PaCo and BCL on CIFAR-100-LT.
>
> **There are quite a few hyper-parameters, like temperature, ,  and  in Eq. 5. Though the authors provide some hyper-parameter study on CIFAR-100-LT, I still wonder how the hyper-parameters change across datasets? Do I need to re-adjust these hyper-parameters with lots of efforts for a new dataset to achieve good performance?**
>
> Thanks for your suggestion. We have provided the detailed experiment hyperparameters for different benchmark datasets on Table 20.
>
> **In Table 1, it seems these contrastive methods are implemented on different code base. To verify the effectiveness of SBCL over the SCL baseline, it is better to use your own implementation on SCL.**
>
> Thanks for your suggestion.
> We have re-implemented these important SCL baselines and reported the accuracy of different disjoint subsets and entire dataset in Table 1.
>
>
> **There are quite a few works trying to re-balance the supervised contrastive loss for long-tailed recognition. What if we instead use non-contrastive methods like BYOL? For non-contrastive methods, the subclass-balancing problem naturally disappears.**
>
> Thanks for your suggestions! We have added the results of BYOL in Table 1.  From the results, we can see that BYOL underperforms SBCL.
>
> [1] Parametric contrastive learning, ICCV 2021.
>
> [2] Balanced Contrastive Learning for Long-Tailed Visual Recognition, CVPR 2022.

---

### Official Review · Reviewer_S8iv · 2022-10-23

**Confidence:** 4
**Correctness:** 2
**Technical Novelty And Significance:** 2
**Empirical Novelty And Significance:** 2
**Recommendation:** 5

**Clarity, Quality, Novelty And Reproducibility:**

The clarity can be improved. Quality of the paper is okay. Novelty is below average. It is unclear how easy it is to reproduce the results (code is not provided).



**Details Of Ethics Concerns:**

No ethics issues as I am aware.

**Strength And Weaknesses:**

Strength:
- Clustering head classes into subgroups for group-level balanced learning is interesting in the context of long-tailed recognition.
- The study of long-tailed pretraining for downstream tasks is interesting.


Below are weaknesses.

For LTR, it is misleading to state that "the major challenge (of LTR) is the discrepancy between the imbalanced training data distribution and the balanced test set." In the real world, testing data should also follow the same long-tailed destruction, but the requirement or evaluation metric is "class balanced", i.e., it requires recognizing all imbalanced classes equally well.

While the paper states that " it might be wise to break down the head classes into multiple semantically coherent subclasses," it should discuss whether the clustering groups correlate to the "semantically coherent subclasses". In other words, how to guarantee the clusters are aware of subclasses?

Algorithm 1 describes a heuristic method about clustering data of a head class into equal-size subgroups. However, would subgroups produced later be noisy (as they will have large distances to their mean features) compared to those generated in early steps? Can authors provide more analyses or visualizations?

Table 2: While SBCL achieves better performance than the compared methods, [R1] performs much better by tuning weight decay in training. [R1] finds that regularizing network weights (towards more balanced norms in network filters) is crucial for LTR, and doing so leads to state-of-the-art. Therefore, it is unclear why SBCL improves LTR (slightly): whether it is due to balanced subgroups of head classes or whether it helps learn balanced weights in networks. Authors should provide more analyses.

[R1] Alshammari, et al., "Long-Tailed Recognition via Weight Balancing", CVPR 2022


Section 4.3 and Table 3: Can authors list results by "training from scratch" (i.e., without pretraining), and "freezing backbone" (i.e., not finetuning the whole pretrained network). The two baselines are important as comparisons can clearly show how much imbalanced/balanced pretraining helps the downstream detection task.

Table 5: It is confusing to compare the absolute distances of intra- and inter-class distances. Instead, a ratio between intra- and inter-class distances might better show how well the per-class examples are separated in the feature space. If computing such a ratio, SBCL seems to perform worse than the other methods. Can authors discuss this point?

The paper does not show how many epochs/iterations needed for convergence by SupCon and SBCL, and for SBCL's warm-up learning. Can authors provide a learning curve, e.g., loss vs. iterations/epochs, accuracy vs. iterations/epochs?


typos:
- "a instance" in Section 3
- "a overly-loose/dense cluster"

**Summary Of The Paper:**

This paper focuses on improving supervised contrastive learning to learn better features over class-imbalanced dataset and to achieve better long-tailed recognition (LTR). The paper argues that, in the literature of LTR, typical class-balancing techniques such a data reweighting, resampling and supervised contrastive learning (SupCon) introduce biases as data instances are imbalanced although classes are balanced in training. Motivated by this, the paper proposes a method called "subclass-balancing contrastive learning (SBCL)", which clusters training data in balanced subgroups within (head) classes such that subgroups and tail classes are balanced in size. SBCL samples positive pairs of data within subgroups or tail classes in SupCon. It reports improved LTR performance in experiments.

**Summary Of The Review:**

In the context of long-tailed recognition, the paper proposes a method to cluster head classes into small subgroups such that subgroups and tail classes have roughly equal number of training data, allowing for more balanced learning of feature representations using Supervised Contrastive Learning. However, the paper lacks justifications for some design choices and results, and misses citations and lacks analyses. Therefore, the paper is rated as "marginally below the acceptance threshold".

---

> ### Author Response · Authors · 2022-11-15
> **Response to Reviewer S8iv**
>
> **For LTR, it is misleading to state that "the major challenge (of LTR) is the discrepancy between the imbalanced training data distribution and the balanced test set." In the real world, testing data should also follow the same long-tailed destruction, but the requirement or evaluation metric is "class balanced", i.e., it requires recognizing all imbalanced classes equally well.**
>
> We have modified this sentence to be ’the major challenge is to require the model recognizing all classes equally well’ in the revised paper.
>
> **While the paper states that " it might be wise to break down the head classes into multiple semantically coherent subclasses," it should discuss whether the clustering groups correlate to the "semantically coherent subclasses". In other words, how to guarantee the clusters are aware of subclasses?**
>
> We have visualized the resultant clusters of different head class of ImageNet-LT in Figure 6.
> As Figure 6 shows, semantically similar samples are assigned to the same subclass, which indicates the capability of SBCL to discover semantically coherent subclasses.
> Also the idea of using clustering to find semantically coherent subclasses are also explored in previous work [1-2].
>
> **Algorithm 1 describes a heuristic method about clustering data of a head class into equal-size subgroups. However, would subgroups produced later be noisy (as they will have large distances to their mean features) compared to those generated in early steps? Can authors provide more analyses or visualizations?**
>
> We think that as the training proceeds, the model indeed learns better representations, which would produce less noisy clusters, while in the early stage of training, the randomly initialized model cannot produce distinguishable representations and we need to warm up the model. And we empirically show that the warm-up could boost the performance as in Table 4.
>
>
> **Table 2: While SBCL achieves better performance than the compared methods, [3] performs much better by tuning weight decay in training. [3] finds that regularizing network weights (towards more balanced norms in network filters) is crucial for LTR, and doing so leads to state-of-the-art. Therefore, it is unclear why SBCL improves LTR (slightly): whether it is due to balanced subgroups of head classes or whether it helps learn balanced weights in networks. Authors should provide more analyses.**
>
> We have visualized the per-class weight norm of a linear classifier trained on top of features learned by SBCL in different training stages in Figure 4.
> As training proceeds, we could see that the weight norm of each class becomes more equal even though when training the linear classifier, a plain cross-entropy loss is used. This partially demonstrates the effectiveness of SBCL in improving the downstream tasks by learning better representations, and we think by balancing the subclasses, the learned feature could help learn balanced classifier weights in later stage.
>
> **Section 4.3 and Table 3: Can authors list results by "training from scratch" (i.e., without pretraining), and "freezing backbone" (i.e., not finetuning the whole pretrained network). The two baselines are important as comparisons can clearly show how much imbalanced/balanced pretraining helps the downstream detection task.**
>
> We have added the experiment results of "training from scratch" in Table 3 and 12.
> For the object detection and instance segmentation visual tasks, we followed previous work (eg, MoCo and KCL ) to finetune the pretrained model, and we found the performance of freezing the pre-trained backbone was quite worse, which was mainly because of the large gap between the pretraining and the target tasks, so we did not include the results.
>
> **Table 5: It is confusing to compare the absolute distances of intra- and inter-class distances. Instead, a ratio between intra- and inter-class distances might better show how well the per-class examples are separated in the feature space. If computing such a ratio, SBCL seems to perform worse than the other methods. Can authors discuss this point?**
>
> Please find the response to this question in **Response to common questions**.
>
> **The paper does not show how many epochs/iterations needed for convergence by SupCon and SBCL, and for SBCL's warm-up learning. Can authors provide a learning curve, e.g., loss vs. iterations/epochs, accuracy vs. iterations/epochs?**
>
> In Figure 5, we have analyzed the effect of training epochs on SBCL.   As Figure 5 shows,  SBCL converges after 800 epochs.
>
>
> [1] ''Prototypical contrastive learning of unsupervised representations'',  ICLR 2021
>
> [2] ''Hcsc: Hierarchical contrastive selective coding'', CVPR 2022
>
> [3] "Long-Tailed Recognition via Weight Balancing", CVPR 2022

---

> > ### Comment · Reviewer_S8iv · 2022-11-17
> > **acknowledging the rebuttal**
> >
> > Thanks for the response. I have read the rebuttal and other reviewers' comments. In general, I am not convinced by authors' response. For example, Figure 6 is hard to follow that visualizes subclasses generated by the proposed method. Is there a meaning of these images' placement in the 2D plane? Moreover, I suggested adding results of training from scratch and freezing backbone in Table 1 for comprehensive understanding, but the authors did not want to add them just because the numbers are low; this is quite confusing.
> >
> > I hope the authors incorporate all reviewers' comment in the future submission.

---

> > > ### Author Response · Authors · 2022-11-19
> > > **Thanks for follow-up feedback**
> > >
> > > Thanks for your feedback.
> > > Figure 6 visualizes the different subclasses of the same class, and we have modified the figures in the latest revised paper to make it clear.
> > >
> > > Following you suggestions, We have added the results of freezing the backbone on the downstream detection task in Table 17. Our method still outperformed other baselines.
> > > And the results of training from scratch have added in Table 3 and Table 12 for comprehensive understanding.

---

### Official Review · Reviewer_QXy7 · 2022-10-23

**Confidence:** 5
**Correctness:** 2
**Technical Novelty And Significance:** 2
**Empirical Novelty And Significance:** 2
**Recommendation:** 5

**Clarity, Quality, Novelty And Reproducibility:**

- Clarity: The paper is mostly clearly written
- Quality: The quality in terms of presentation is good, but the quality of evaluation is not sufficient as I pointed out above.
- Novelty: The idea of introducing sub-class contrastive learning into long-tail problems has certain novelty.
- Reproducibility: The procedure of the proposed method is generally well described, but the initialization of the cluster centers in Algorithm 1 is not explicitly explained. Although I believe that basically important information for reproducibility is given in the paper including the appendix, readers may practically face issues since the code is not provided.


**Strength And Weaknesses:**

## Strength
1. The proposed method is evaluated on wide variety of datasets. Especially the evaluation on other tasks than image classification, i.e., object detection and semantic segmentation, is given.
1. The effectiveness of the proposed method is shown in the experiment at least to some extent.
1. The paper generally reads well

## Weakness
1. The comparison with stronger baselines such as [A1-3] is missing. The reported accuracies in these works are actually higher than the proposed method.
1. According to Table 4, the performance of the proposed method on CIFAR100-LT becomes 43.7 if the dynamic temperature is not applied. This performance is lower than TSC. I am afraid this suggests that it is the dynamic temperature that makes the paper perform better than other contrastive learning methods. To show the effectiveness of the proposed idea, it is necessary to show the results of other contrastive learning methods such as SCL, KCL, and TSC with the dynamic parameters.
1. I am afraid the performance of the proposed method is strongly affected by the batch size since the distribution in each mini batch is important in calculating the loss in Eq. (4). It would be nice if the authors could provide the analysis on this point.
1. Table 5: “(i) the inter-class distances of SBCL is larger than the other methods, which implies that SBCL can push different classes far away from each other, and thus clear the decision boundaries between classes; ” It is true that the inter-class distances of SBCL is larger than those of the other methods, but the intra-class distance of SBCL is also larger. Actually, the intra-class distances of other methods are much smaller than those of the proposed method, which means that the other methods have favorable characteristics.
1. P8 “We attribute that to feature space dominated by the head class.” Please rephrase and elaborate more so that the claim of this sentence becomes clearer.

[A1] Zhong+, “Improving Calibration for Long-Tailed Recognition”, CVPR 2021
[A2] Cui+, “Parametric Contrastive Learning”, ICCV 2021
[A3] Zhu+, “Balanced Contrastive Learning for Long-Tailed Visual Recognition”, CVPR 2022

Minor points.
1. P2, (c) “… and instant segmentation” -> “instance segmentation”
1. Eq (2): $\tilde{V}_i^k$ -> $\tilde{P}_i^k$?
1. The second line in Section 3 $|V^+_{i, k}|$ does not appear in equation (2)
1. P8 “successful captured” -> ““successfully captured””
1. P9 “well-haved” What does this mean?
1. P9: “the table 4” -> “Table 4”


**Summary Of The Paper:**

This paper proposes a new method called subclass balancing contrastive learning for long-tailed problems. The key idea is to divide head classes into subclasses to balance the number of instances in each subclass. The proposed method is evaluated on widely used image classification datasets as well as some additional vision tasks such as object detection and semantic segmentation.

**Summary Of The Review:**

I think the motivation of the paper is sound and the proposed method is somehow reasonable. However, I have doubt on the effectiveness of the proposed method since I do not think the current experiment is not sufficient for supporting it. As such I do not think this paper is ready for publication. I would like to hear the feedback from the authors on the points I listed in the weakness.

### after rebuttal
I became more positive about the paper through the rebuttal and I increased my score accordingly, but I still have some concerns on the effectiveness of the proposed method.

---

> ### Author Response · Authors · 2022-11-15
> **Response to Reviewer QXy7**
>
> **The comparison with stronger baselines such as [1-3] is missing. The reported accuracies in these works are actually higher than the proposed method.**
>
> Thanks for suggesting these baselines! We think these methods focus more on training a better classifier, while our focus is the representation learning part of a two-stage pipeline.
> We have added experiments of comparing SBCL with the suggested methods in Table 14 15 and 16. The results in Table 14  and 16 suggest that using a model pretrained with SBCL could boost the performance of these baselines.
> In Table 15,  we replaced the supervised contrastive learning in PaCo and BCL with SBCL, and it improves over the PaCo and BCL.
> In fact, these results suggest that an interesting future work in long-tail recognition would be to find the best practice that is a combination of multiple existing techniques, and we think SBCL could be one of the candidates since it can be used to improve the performance of other methods.
>
>
> **According to Table 4, the performance of the proposed method on CIFAR100-LT becomes 43.7 if the dynamic temperature is not applied. This performance is lower than TSC. I am afraid this suggests that it is the dynamic temperature that makes the paper perform better than other contrastive learning methods. To show the effectiveness of the proposed idea, it is necessary to show the results of other contrastive learning methods such as SCL, KCL, and TSC with the dynamic parameters.**
>
>  In Table 9, we have applied the dynamic temperature on the second term of TSC and it improves the performance of TSC.
> However,  TSC with dynamic temperature still underperforms our method, and we think it is because randomly draws equal number of samples in same class would negatively affect the efficacy of the dynamic temperature.
>
>
> **The performance of the proposed method is strongly affected by the batch size since the distribution in each mini batch is important in calculating the loss in Eq. (4). It would be nice if the authors could provide the analysis on this point.**
>
> In our revision paper , we have provided an analysis of the relationship between batch size and the top-1 accuracy of classification on SBCL in Figure 5.
>
> **Table 5:  (i) the inter-class distances of SBCL is larger than the other methods, which implies that SBCL can push different classes far away from each other, and thus clear the decision boundaries between classes; ” It is true that the inter-class distances of SBCL is larger than those of the other methods, but the intra-class distance of SBCL is also larger. Actually, the intra-class distances of other methods are much smaller than those of the proposed method, which means that the other methods have favorable characteristics.**
>
> Please find the response to this question in **Response to common questions**.
>
> **P8 “We attribute that to feature space dominated by the head class.” Please rephrase and elaborate more so that the claim of this sentence becomes clearer.**
>
> We have edited this Section 4.4 in the revised paper.
>  KCL [4] and TSC [5] have relatively more discrepant intra-class distance across many/medium/few and the head classes have lower intra-class distance, which indicates that the head classes actually form more dense clusters in feature space than tail classes; this is counter-intuitive because head classes usually contain rich semantic information. The underlying reason for this phenomenon could be that the training is biased towards head classes and they have more chances to be updated and pushed together.
>
>
> [1]  “Improving Calibration for Long-Tailed Recognition”, CVPR 2021
>
> [2]  “Parametric Contrastive Learning”, ICCV 2021
>
> [3]  “Balanced Contrastive Learning for Long-Tailed Visual Recognition”, CVPR 2022
>
> [4] ''Exploring balanced feature spaces for representation learning'', ICLR 2020
>
> [5] ''Targeted supervised contrastive learning for long-tailed recognition'', CVPR 2022

---

> > ### Comment · Reviewer_QXy7 · 2022-11-18
> > **Thank you for the feedback**
> >
> > I appreciate the authors' feedback.
> > I think the paper becomes much more convincing by the intensive efforts of the authors.
> > For example, the comparison and combination with stronger baselines that I raised made the paper stronger.
> >
> > On the other hand, I am not fully convinced by the response.
> > Especially the discussion on the inter/intra class distance is not that convincing as reviewer S8iv also pointed out.
> > In addition, Fig.5(b) indicates that the performance of the proposed method greatly decreases if the batch size is smaller than 500, which is the drawback of the method.
> >
> > Overall, I became more positive about the paper through the rebuttal and I increased my score as such, but still leaning to the rejection.

---

> > > ### Author Response · Authors · 2022-11-19
> > > **Thanks for follow-up feedback**
> > >
> > > Thank you for your follow-up feedback and we are really appreciated for your recognition of our work.
> > >
> > > In our latest revised paper, we have added the hyperparameter analysis of TSC in Figure 5 as well. From the Figure 5, we can see that the performance of TSC also significantly decreases when batch size is set to 256 on CIFAR-100-LT, because SBCL and TSC are built on top of SimCLR-based supervised contrastive learning, which is sensitive to batch size. However, SBCL outperforms TSC across different batch size, which indicates the superiority of SBCL over other SimCLR [1]-based baseline.
> > >
> > > We apologize that our response regarding intra-/inter-class distance above caused some confusion. The main takeaway of our intra-/inter-class distance analysis as we described in the paper is the SBCL could achieve more balanced intra-class distance and larger inter-class distance than baselines. And what we put in previous response is a speculation of why increased intra-class distance is not harmful, and we welcome any further discussion on that point from both reviewers and other interested readers. To the best of our knowledge, there is no concrete empirical/theorectical evidence existed to indicate the increased intra-class distance is harmful or not. And please note that we do not claim the increased intra-class disatnce as an advantage of our method in the paper.
> > >
> > > [1] ''A Simple Framework for Contrastive Learning of Visual Representations'', ICML 2020

---

### Official Review · Reviewer_8khH · 2022-10-29

**Confidence:** 4
**Correctness:** 3
**Technical Novelty And Significance:** 2
**Empirical Novelty And Significance:** 3
**Recommendation:** 6

**Clarity, Quality, Novelty And Reproducibility:**

**Clarity and Novelty**
The paper is clearly written and seems novel.

**Reproducibility**
I am not sure if the paper has sufficient training details to confirm reproducibility. However, it is mentioned that the training process is same as TSC (Li et al., 2022).

**Strength And Weaknesses:**

**Strengths**

S1: Extensive set of experiments on several vision tasks show SBCL does better than other representation learning methods for learning on long-tail distribution. Many ablations were also included to show several design choices made.

S2: I think the idea of clustering the head class(es) into sub-classes and using the cluster ids as labels is novel.

**Weaknesses**

W1: While there are extensive sets of experiments, the quantitative results (Table 1-5) do not come with standard errors/deviations. The experiments were done w/ 5 random seeds, then I wonder why have the standard errors not been included in the table? Understanding how reliable these improvements seems like an important thing to report.

W2: There is no discussion on the convergence of the model, and the number of training steps/epochs, loss trajectory or learning rate used. Since SBCL uses warm-up and adaptive clustering which may lead to a jump in the loss fn, studying these properties seems highly relevant. It is unclear if the number of training steps/epoch used for the training the feature extractor is the same for all the baselines and the proposed SBCL.

Other weakness/questions:
- I wonder why the disjoint subset analysis was not shown for the CIFAR-100 LT dataset?
- The conclusion from the analysis in section 4.4 lacks clarity. What is the conclusion from this analysis? What is the significance of the finding: “the intra-class distance of SBCL is almost invariant with the decreasing of group split”. Why the increase in the intra-class distance for KCL & TSC is attributed to the head class? The discussion in this section needs some work.
- Was the negative sampling bias corrected when computing the loss function? While SBCL balances the label distribution (via cluster labels for head instances), it is unclear that this would mitigate the sampling bias in selecting negatives. Typically, models use log(P(x)) correction [1] when using in-batch softmax loss to prevent the head instances to be overly treated as negatives than the tail instances.Having a baselines that uses this logit adjustment would be a definite plus.

[1] Menon et al. (2021), Long-tail learning via logit adjustment, (ICLR 2021).


**Summary Of The Paper:**

This paper proposes a method for contrastive learning on a long-tailed training dataset. The proposed method is called Subclass Balancing Contrastive Learner (SBCL), which uses clustering within the head classes to determine fine-grained subclasses of size similar to the tail clusters. The new subclass labels for the head instances are used w/ tail classes to perform supervised contrastive learning. The paper contains extensive experiments over several vision benchmarks, including classification, object recognition and instance segmentation tasks. In almost all cases, SBCL performs better than several other baselines.


**Summary Of The Review:**

(Please see the strengths and weaknesses above.) I would appreciate if the authors respond to some of my questions and concerns.

---

> ### Author Response · Authors · 2022-11-15
> **Response to Reviewer 8khH**
>
> **While there are extensive sets of experiments, the quantitative results (Table 1-5) do not come with standard errors/deviations. The experiments were done w/ 5 random seeds, then I wonder why have the standard errors not been included in the table? Understanding how reliable these improvements seems like an important thing to report.**
>
> Thanks for your advice to make our paper more precise. In the revised paper, we  have added the standard deviations
> of results with five random seeds.
>
>
> **There is no discussion on the convergence of the model, and the number of training steps/epochs, loss trajectory or learning rate used. Since SBCL uses warm-up and adaptive clustering which may lead to a jump in the loss fn, studying these properties seems highly relevant. It is unclear if the number of training steps/epoch used for the training the feature extractor is the same for all the baselines and the proposed SBCL.**
>
> Thanks for your advice.
> The impact of different training epochs and learning rate on SBCL has been analysed on Figure 5.
> The purpose of warm-up stage is to avoid noisy clustering results caused by uninformative representation at the early stage of the training, and we do not observe that warm-up stage/adaptive clustering affects the convergence of the model during training.
> Table 20 has provided the detailed hyperparaemeter for different benchmark datasets trained by SBCL, and the number of training epoch for all the SCL baselines is the same as the proposed SBCL.
>
>
> **I wonder why the disjoint subset analysis was not shown for the CIFAR-100 LT dataset?**
>
> We have added the results of disjoint subsets on the CIFAR-100-LT dataset with the standard errors in Table 2.
>
>
> **The conclusion from the analysis in section 4.4 lacks clarity. What is the conclusion from this analysis? What is the significance of the finding: “the intra-class distance of SBCL is almost invariant with the decreasing of group split”. Why the increase in the intra-class distance for KCL & TSC is attributed to the head class? The discussion in this section needs some work.**
>
> Please find the response to this question in **Response to common questions**.
>
> **Was the negative sampling bias corrected when computing the loss function? While SBCL balances the label distribution (via cluster labels for head instances), it is unclear that this would mitigate the sampling bias in selecting negatives. Typically, models use log(P(x)) correction [1] when using in-batch softmax loss to prevent the head instances to be overly treated as negatives than the tail instances.Having a baselines that uses this logit adjustment would be a definite plus.**
>
> Thanks for your great question.
> We have added an additional experiment (Table 6) to answer this question.
> We followed the suggestion to let the first term of SBCL use instances in different classes as negatives.
> According to the results of the Table 6,  this decreases the model performance.
> At the same time, Figure 3  shows that SBCL does not improve the recognition of the tail analogy at the expense of the accuracy of the head category.
> The logit adjustment mentioned is orthogonal to our method and can be applied on any SCL-based method, so we do not include it for a fair comparison.
>
> [1] ''Long-tail learning via logit adjustment'', ICLR 2021

---

### Author Response · Authors · 2022-11-15
**Summary of revision**

Dear Reviewers and Area Chair,

We thank all of your efforts and valuable time on our paper.
Thank reviewers for affirming the motivation of our work, technical contribution, paper organization, and downstream tasks experiments.
We have updated a new version of the article to address  the reviewers' questions/concerns, rephrase some of the sentences/paragraphs to clarify, and fixed typos.
Next, we introduce the revised parts of the article according to all reviewer's comments.
The revised contents are highlighted in blue for ease of reading and code is provided in supplementary material.
**We are more than happy to answer any futher questions.**

### Writing

We have fixed some typos and ambiguous statements in our revised paper.

#### Respones to Reviewer S8iv:

(1): In Section 2 of Page 2 ,  the sentence of 'the major challenge (of LTR) is the discrepancy between the imbalanced training data distribution and the balanced test set.' is rephrased as 'the major challenge is to require model recognizing
all classes equally well.'.


(2): In Section 3,  'a instnance'  and 'a overly-loose/dense cluster' are changed as 'an instance' and 'an overly-loose/dense cluster'.


#### Respones to Reviewer QXy7:

(1): In Section 1 of Page 2, we correct '… and instant segmentation' as 'instance segmentation' in introduction section.

(2): In Section 2 and 3 of Page 3, we change '$\tilde{V_{i}^{k}}$' to '$\tilde{P_{i}^{k}}$' in  Eq.(2). The original character of $|V_{i,k}^{+}|$ is corrected as $|P_{i}^{k}|$.

(3): In Section 4.4 of Page 8,  the new sentence is '....the two-layer class hierarchy are successfully captured'

(4): In the Section 4.5 of Page 9, 'well-haved' and 'the table 4' are converted into 'well-trained' and  'Table 4'.

### Experiment

We have refined and added a bunch of additional experiments based on the reviewers' comments.

(1) [suggested by Reviewer 8khH]: We have added the standard error for our main downstream tasks with five different random seeds on Table 1-5.


(2) We have visualized per-class accuracy of both SBCL and SCL on CIFAR-100-LT in Figure 3.


(3) [suggested by Reviewer S8iv]: We have visualized the per-class weight norm of a linear classifier trained on top of features
learned by SBCL in different training stages in Figure 4.


(4) [suggested by Reviewer 8khH]: We experimented with different negative sample selections for SBCL in Table 6 to resolve the question of whether the head instances are be overly treated as negatives.


(5) [suggested by Reviewer QXy7]: We incorporated dynamic technique in TSC on Table 9.

(6) [suggested by Reviewer 8khH, QXy7, and S8iv]: we have added the hyperparameter (learning rate, training epochs and batch size) analysis on SBCL in Figure 5.

(7) [suggested by Reviewer QXy7 and aZoT] We have added the suggested baselines (BYOL [1], PaCo [2], BCL [3], and MiSLAS [4]) in Table  2, 14 , 15 and 16.

(8) [suggested by Reviewer S8iv]: We have visualised the results of clusters from different head classes generated by SBCL in Figure 6.

(9) [suggested by Reviewer 8khH and aZoT]: We have provided the detailed experiment hyperparameters for different benchmark datasets on Table 20.

(10)[suggested by Reviewer S8iv]:  We have added the results of  training from scratch and freezing backbone in Table 3, 12 and 17.

Best regards.

[1] 'Bootstrap Your Own Latent  A New Approach to Self-Supervised Learning', NIPS2020

[2]  “Parametric Contrastive Learning”, ICCV 2021

[3] “Balanced Contrastive Learning for Long-Tailed Visual Recognition”, CVPR 2022

[4] “Improving Calibration for Long-Tailed Recognition”, CVPR 2021

---

### Author Response · Authors · 2022-11-15
**Response to common questions**

There is one question shared by multiple reviewers: the detailed analysis of the intra/inter-class distance. We have added more analysis on the intra/inter-class distance in the Section 4.4 and answer this question in details as below.

First, the intra-class distance of SBCL is less sensitive to the size of the training data in each class than baselines, which indeed reflects that the SBCL is able to reduce the bias towards the head classes by assigning more equal volume of space to each class.

Second, regarding the larger intra-class distance than that of other methods, we think it is an advantage of SBCL rather than caveat, because as long as we maintain large inter-class distance, larger intra-class distance indicates that we utilize the space of learned representation more effectively and avoid overly-dense classes in the space, which would make the model less robust to the permutation of the data.

In terms of the ratio of intra/inter-class distance, we argue that the distinguishability of the learned feature is mainly determined by the inter-class distance, and our method provides larger inter-class distance than baselines. But it is not necessarily true that the smaller ratio of intra/inter-class distance is better, because a smaller ratio of intra/inter-class distance may lead to overly-dense class.

In addition, previous work [1] has shown that for contrastive learning, the uniformity is a key property to preserve as much information of individual data as possible and the relatively larger intra-class distance provided by SBCL indicates its learned space is more uniform than other methods.
Conceptually, a dense class in feature space actually means that a lot of data points are mapped into a small volume of the feature space, losing a lot of information of individual data points, which may result in same sort of overfitting- only a few patterns of the class are preserved and therefore could hurt the performance on downstream tasks.

[1] ''Understanding the behaviour of contrastive loss'' , CVPR 2021

---

### Decision · Program_Chairs · 2023-01-20

**Decision:**

Reject

**Justification For Why Not Higher Score:**

Improvements over baselines are in some cases small and not completely clear as to statistical significance; some remaining concerns on technical novelty; and discussion on intra- and inter-class distance was found to be not very convincing, due to speculative nature.

**Justification For Why Not Lower Score:**

N/A

**Metareview: Summary, Strengths And Weaknesses:**

The paper proposes their SBCL approach for contrastive learning in the long-tailed setting. They use clustering within the head classes to break them into fine-grained subclasses of similar size to the tail classes. They then consider losses both at the subclass and at the class level. In addition, a dynamic temperature approach to assign appropriate temperature parameters for different class sizes is proposed.

In general, reviewers find the paper to be clearly presented, and found the general approach of enforcing instance and class balance by breaking head classes into subclasses to be interesting and well motivated. However, there are a number of main issues raised by reviewers. Some reviewers felt that the performance improvements were not convincing enough, and also suggested adding some stronger baselines. Moreover, most reviewers felt that the analysis of intra-class distance vs inter-class distance (e.g. table 5) was not very convincing - considering that both the intra- and inter-class distances of the proposed method are higher than those of the baselines, and the ratio is not seemingly favorable either. WIth regard to novelty, reviewers generally felt that the paper has a certain novelty in its use of subclasses, though some felt that its technical novelty was somewhat marginal.

In the rebuttal stage, the author response made a number of improvements to the paper, which helped to address some of the reviewer concerns, such as adding hyperparameter information, standard errors, other additional experiments + writing improvements etc - I thank the author for their efforts in addressing a range of concerns and improving the paper, including the addition of stronger baselines including BYOL/PaCo/BCL/MiSLAS.

As this was a borderline paper, a virtual meeting was conducted. Reviewers felt that several of the concerns were satisfactorily addressed (hyperparameters, etc). With regard to the new stronger baselines, reviewers felt that this was an improvement, though still considered that the new improvements (e.g. table 14/15) were sometimes small and not completely clear as to their statistical significance. However, regarding the analysis of intra- and inter-class distance, reviewers still felt that the updated results were not fully convincing, particularly as they seem overly speculative (e.g. in terms of the meaning of intra- and inter-class distance). Reviewers further suggested improvements to the subclass visualization, such as using visualization techniques similar to t-sne to also visualize the entire latent space (as an add-on to the existing visualizations, which focus on a small number of classes).

In the end, reviewers and AC agree that while the work has promising merits, due to these issues, in addition to some remaining concerns on technical novelty, the work is not yet ready for publication at ICLR. The reviews offer a number of helpful suggestions for improvement, so I encourage the authors to continue improving the paper based on the reviews for future submissions.

**Summary Of Ac-Reviewer Meeting:**

Several concerns like hyperparameter information, standard errors, etc were satisfactorily addressed. Reviewers and AC agreed that the stronger baselines were an improvement, but still considered that the new improvements were sometimes small and not completely clear as to their statistical significance. For intra- and inter-class distance, the updated results were seen as not very convincing, particularly due to the speculative nature. Reviewers further suggested improvements to the subclass visualization. In the end, these remaining issues (along with some remaining concerns on technical novelty) were seen as significant considerations for the final decision.